# Phytochemical Profiling, In Vitro Biological Activities, and In-Silico Studies of *Ficus vasta Forssk.*: An Unexplored Plant

**DOI:** 10.3390/antibiotics11091155

**Published:** 2022-08-26

**Authors:** Hanan Y. Aati, Mariyam Anwar, Jawaher Al-Qahtani, Areej Al-Taweel, Kashif-ur-Rehman Khan, Sultan Aati, Faisal Usman, Bilal Ahmad Ghalloo, Hafiz Muhammad Asif, Jafir Hussain Shirazi, Aliza Abbasi

**Affiliations:** 1Department of Pharmacognosy, College of Pharmacy, King Saud University, Riyadh 11495, Saudi Arabia; 2Department of Pharmaceutical Chemistry, Faculty of Pharmacy, The Islamia University of Bahawalpur, Bahawalpur 63100, Pakistan; 3UWA, University of Western Australia, Nedland, WA 6009, Australia; 4Department of Pharmaceutics, Faculty of Pharmacy, Bahauddin Zakariya University, Multan 60000, Pakistan; 5Faculty of Medicine and Allied Health Sciences, University College of Conventional Medicine, The Islamia University of Bahawalpur, Bahawalpur 63100, Pakistan; 6Department of Pharmaceutics, Faculty of Pharmacy, The Islamia University of Bahawalpur, Bahawalpur 63100, Pakistan

**Keywords:** *Ficus vasta*, antibacterial, anti-viral, antifungal, thrombolytic, HepG2 cell line, GCMS, molecular docking, ADMET

## Abstract

*Ficus vasta* Forssk. (Moraceae family) is an important medicinal plant that has not been previously investigated for its phytochemical and biological potential. Phytochemical screening, total bioactive content, and GCMS analysis were used to determine its phytoconstituents profile. Antioxidant, antibacterial, antifungal, anti-viral, cytotoxicity, thrombolytic, and enzyme inhibition activities were examined for biological evaluation. The plant extract exhibited the maximum total phenolic (89.47 ± 3.21 mg GAE/g) and total flavonoid contents (129.2 ± 4.14 mg QE/g), which may be related to the higher antioxidant potential of the extract. The extract showed strong α-amylase (IC_50_ 5 ± 0.21 µg/mL) and α-glucosidase inhibition activity (IC_50_ 5 ± 0.32 µg/mL). Significant results were observed in the case of antibacterial, antifungal, and anti-viral activities. The *F. vasta* extract inhibited the growth of HepG2 cells in a dose-dependent manner. The GCMS analysis of the extract provided the preliminary identification of 28 phytocompounds. In addition, the compounds identified by GCMS were subjected to in silico molecular docking analysis in order to identify any interactions between the compounds and enzymes (α-amylase and α-glucosidase). After that, the best-docked compounds were subjected to ADMET studies which provide information on pharmacokinetics, drug-likeness, physicochemical properties, and toxicity. The present study highlighted that the ethanol extract of *F. vasta* has antidiabetic, antimicrobial, anti-viral, and anti-cancer potentials that can be further explored for novel drug development.

## 1. Introduction

Medicinal plants have played a major role as source of lead compounds that may be used for a variety of medicinal and pharmacological activities [1]. These plants are commonly used to treat various diseases including asthma, diabetes, respiratory, gastrointestinal diseases, skin disorders, urinary, cardiovascular, and hepatic diseases [2]. Recently, due to the wide variety of secondary metabolites identified in medicinal plants, there has been a greater focus on research which can help to obtain lead compounds from medicinal plants [3]. Some countries have started botanical drug research projects to investigate phyto-pharmaceuticals due to the enormous therapeutic potential of plants [4]. Diabetes is a metabolic disease that is caused by hyperglycemia due to the deficiency of insulin secretion or insulin action (or both in some cases) [5]. As the incidence of diabetes continues to rise, there is an increasing interest in plants with strong antidiabetic properties. These plants could provide a breakthrough in the fight against this disease [6]. There are many mechanisms by which antidiabetic drugs can help to regulate blood sugar levels. These include suppressing hepatic glucose production, improving the sensitivity of insulin receptors, stimulating insulin secretion, and increasing peripheral glucose uptake. In addition, antidiabetic drugs can also delay digestion and absorption of carbohydrates from the gut, helping to control post-meal blood sugar levels [7]. Recent research suggests that α-amylase and α-glucosidase inhibitors help to control blood sugar levels post-meal by delaying digestion [5]. α-Glucosidase and α-amylase inhibitors such as acarbose and miglitol inhibited the absorption of carbohydrates from the intestine [8]. The traditional medicinal plants are rich source of chemical constituents and phytotherapeutic preparations for the development of antidiabetic drugs [9].

Liver cancer is the second main cause of death globally [10]. Hepatocellular carcinoma causes around 90% of all primary liver cancers [11], and chemotherapy, radiation treatment, or both may be used to treat advanced liver cancer [12]. Many studies have shown that medicinal plants have compounds that can slow cancer progression [13]. Antioxidants are compounds that act as scavengers of free radicals created by the body’s cells due to various metabolic reactions and external factors. These free radicals are unstable and one of the most common causes of cancer in the human body, they cause serious damage to the proteins, lipids, DNA, and RNA [12].

Infectious diseases are a main cause of death and illness in developing countries. They are a serious public health problem that can have a devastating impact on communities [14]. There are different types of pathogenic microbes that can cause illness, including fungi, bacteria, parasites, and viruses [15]. Scientists have been prompted to seek for novel compounds with antimicrobial activity from a wide range of sources, including medicinal plants, as a result of the increase cases of multidrug-resistant (MDR) infections [16]. Plant polyphenols are known to have medicinal qualities, including antibacterial, antifungal, and anti-viral activities. This is due to the fact they can induce damage to the bacterial cell membrane, either structurally or functionally. Different studies have revealed that polyphenol-rich plants have antioxidant and antibacterial activities [17]. To effectively combat fungal illnesses, new antifungals must be developed. In this regard, interest in alternative medicinal approaches is expanding, and the use of plant extracts as antimicrobial adjuvants has gotten much attention in recent years [18]. Polyphenols are a promising area in research for developing anti-viral medications. They work in various ways, including preventing virus entrance or affecting virus replication. Polyphenols are widely available and relatively inexpensive to produce, making them an attractive option for medication development [19].

*Ficus* is a genus that belongs to the Moraceae family of woody trees, shrubs, and vines that includes roughly 800 species and 2000 variations. *Ficus* plants have long been used in traditional medicine for their wide range of therapeutic properties. Recent studies have revealed that these plants possess various biological activities, including antioxidant, anti-inflammatory, anti-cancer, antidiabetic, anti-tumor, antiproliferative, antimicrobial, antihelminthic, and hepatoprotective effects [20]. These properties make *Ficus* a promising natural therapy for many different diseases and conditions. *Ficus vasta* Forssk. is a large tree that can grow over 25 m tall. Its leaves are alternate and spirally arranged with an uneven texture. They are almost circular in shape, with an entire margin and a rounded tip, though often with a blunt point. The leaves are usually glabrescent above and below, puberulous or hirsute [21]. These are widespread in dry, arid regions of Africa, such as Sudan, Uganda, Saudi Arabia, Tanzania, and Ethiopia. The leaves and bark can be used to prepare a poultice for tumors [22]. The leaves of *F. vasta* are traditionally used to treat rheumatic diseases, intestinal worms, and muscle pains. Preliminary phytochemical analysis showed that it consists of carbohydrates, flavonoids, tannins, coumarins, and triterpenes [23]. Some phytochemical constituents such as beta-sitosterol, lupeol, stigmasterol, and ursolic acid were isolated and identified from the ariel part of this plant [24]. Moreover, antioxidant and antibacterial activities are investigated in a hydro-methanolic extract of *F. vasta* leaves [25].

The literature review regarding *F. vasta* demonstrated that there are only few data in terms of phytochemical investigations and biological activities as compared to the other members of the genus *Ficus*. Therefore, the goal of this research was to explore the aerial parts of *F. vasta* for its phytochemical and biological activities. The therapeutic potential of *F. vasta* was assessed for the first time through its antioxidant, antibacterial, antifungal, anti-viral, cytotoxicity, and enzyme inhibitory activities. GCMS was used to identify the phytocompounds responsible for these activities. The major phytocompounds identified by GCMS in the ethanolic extract were further studied for in silico studies.

## 2. Results

### 2.1. Phytochemical Analysis

#### 2.1.1. Preliminary Phytochemical Profiling

Phytoconstituents of *F. vasta* extract were analyzed by the phytochemical profiling tests. Table 1. showed that primary and secondary metabolites (i.e., carbohydrates, alkaloids, flavonoids, steroids, glycosides, phenols, tannins, and saponins are present in the ethanolic extract of *F. vasta*).

#### 2.1.2. Total Bioactive Contents

The results of the total phenolic and flavonoid content of *F. vasta* are shown in Table 2. The TPC and TFC of *F. vasta* extract was carried out with the help of regression equation that was obtained from gallic acid and quercetin standard curve. The ethanolic extract of aerial parts of *F. vasta* showed total phenolic content of 89.47 ± 3.21mg GAE/g and total flavonoid contents of 129.2 ± 4.14mg QE/g.

#### 2.1.3. Gas chromatography-Mass Spectroscopy Analysis

In-depth phytochemical identification, the ethanolic extract was further evaluated by GCMS. The GCMS chromatogram of *F. vasta* ethanolic extract showed 28 peaks of compounds. The GCMS chromatogram of compounds was identified by comparing their peak retention time, height (percent), and patterns of mass spectral fragmentation to those of known compounds present in the National Institute of Standards and Technology library (NIST). In GCMS, most of the compounds related to the classes like fatty acids, steroids, vitamins, and esters. Major compounds of GCMS consisting of Stigmasterol, 22,23-dihydro (8.04%), 11-Oxours-12-en-3-yl acetate (6.46%), Phytol (5.13%), n-Hexadecanoic acid (4.27%), Quinic acid (4.07%), 11 14 17-eicosatrienoic acid methyl ester (3.76%), 4-((1E)-3-Hydroxy-1-propenyl)-2-methoxyphenol (3.71%), 2,3-Dihydrobenzofuran (2.17%), and many others minor compounds. The identified compounds from ethanol extract are described in Table 3 and GCMS chromatogram is represented in Figure 1.

### 2.2. Biological Activities

#### 2.2.1. Antioxidant Activities

In this study, the antioxidant activity of *F. vasta* extract was examined by the DPPH, ABTS, FRAP, and CUPRAC, and the results are shown in Table 2. In the case of radical scavenging activities, the DPPH result suggested that the ethanolic extract of aerial parts of *F. vasta* exhibited the highest activity with IC_50_ 1.75 mg/mL. In comparison, the ABTS result showed the activity with IC_50_ 1.63 mg/mL. According to the reducing power assays, the FRAP and CUPRAC results showed the highest activity with IC_50,_ i.e., 1.91 mg/mL, and 1.51 mg/mL, respectively. The results of FRAP and DPPH activities of the extract suggested a direct correlation with the polyphenol content.

#### 2.2.2. Enzyme Inhibition Activities

This study used acarbose as a positive control to determine the enzyme inhibition activity. The α-glucosidase activity result showed that ethanolic extract of *F. vasta* exhibits the excellent α-glucosidase inhibitory effect with IC_50_ 5 ± 0.32 µg/mL compared to the standard acarbose with IC_50_ 2.59 ± 0.11 µg/mL. At the same time, the result of α-amylase activity showed that ethanolic extract of *F. vasta* showed an excellent α-amylase inhibitory effect with IC_50_ 5 ± 0.21 µg/mL compared to the standard acarbose with IC_50_ 41.10 ± 3.21 µg/mL. The results of α-glucosidase and α-amylase inhibition activities were shown in Figure 2.

#### 2.2.3. Antibacterial Activity

The antibacterial potential of *F. vasta* was evaluated using the agar well diffusion method against five positive strains include *Bacillus subtilis*, *Bacillus pumilus*, *Staphylococcus epidermidis*, *Staphylococcus aureus*, *Micrococcus luteus*, and two negative strains including *Escherichia coli*, *and Bordetella bronchiseptica* and results are shown in Table 4 and Appendix A. This antibacterial potential was revealed by inhibition of growth of bacteria which was measeured as zone of inhibition. The results showed that plant extract has a dose-dependent effect on bacteria. Its activity against the bacteria is lowest at low concentrations (25 mg/mL). However, antibacterial activity was observed to be higher for *S. aureus* and *E. coli*, with 22 mm and 24 mm zones of inhibition, respectively, at a concentration of 100 mg/mL.

#### 2.2.4. Antifungal Activity

Table 5 presented the antifungal activity of *F. vasta* extract. The results of this activity showed the maximum antifungal activity of extract was observed against fungal strain *F. avenaceum* (54.94%) followed by *F. brachygibbosum* (48.51%) and *A. niger* (42%).

#### 2.2.5. Anti-Viral Activity

The anti-viral potential against three strains (i.e., avian infectious bronchitis virus (IBV), influenza virus (H9), and Newcastle disease virus (NDV)) was carried out using a haemagglution test method, and the results are tabulated in Table 6. The ethanolic extract of *F. vasta* was more active against the avian infectious bronchitis virus and Newcastle disease virus (IBV and NDV titer 0) while less effective against the influenza virus and its H9 titer was 16.

#### 2.2.6. Thrombolytic Activity

The results of thrombolytic activity of *F. vasta* and Streptokinase are described in Figure 3. The percentage clot lysis of ethanolic extract of *F. vasta* was 47.78 ± 4.21% while streptokinase showed 81.86 ± 6.37% clot lysis.

#### 2.2.7. Hemolytic Activity

Data shown in (Table 7) represented the hemolytic activity of ethanolic extract of *F. vasta*. The hemolytic % value of extract was 6.39 ± 0.45% while the value of standard was 92.27 ± 4.71%. The hemolysis activity of extract was less than 30%, so extract is nontoxic and safe as food. The hemolysis activity can be performed using a wide range of methods, which is one of its major challenges and limitations. This produces a wide range of absolute values in addition to complicates comparisons between studies [57].

#### 2.2.8. Cytotoxicity

MTT activity was used to evaluate the effect of *F. vasta* extract on HepG2 cell growth. Treatment with *F. vasta* extract for 48 h inhibited the growth of cells in a dose-dependent manner. The IC_50_ value for this assay was 0.563 µg/mL. The outcomes of the activity are represented in Figure 4.

### 2.3. In Silico Studies

#### 2.3.1. Molecular Docking

Molecular docking studies were conducted against all compounds that were identified by the GCMS for both α-glucosidase and α-amylase. The docking results suggested that the binding affinity of 7 compounds was higher than the binding affinity of acarbose (standard) in α-glucosidase and α-amylase. Ursa-9(11),12-dien-3-ol was the most active of all of the tested compounds, with a binding affinity of −10.5, which interacted in the active site of α-amylase by forming one H-bond interaction with ASP 197 amino acid residue at a bond distance of 2.87 angstroms. From the 2D structures of ligands with α-amylase, it was clear that beta-amyrin, Campesterol, beta-Sitosterol, and Stigmasterol showed H-bond interactions. In contrast, Olean-12-en-3-ol, acetate, (3beta)- and 11-Oxours-12-en-3-yl acetate did not show any H-bond interactions. The docking results of α-glucosidase showed that beta-amyrin exhibit the highest binding affinity of −8.4 by making H-bond interaction with GLU 113 amino acid residue with a bond distance of 2.78 angstroms. Olean-12-en-3-ol, acetate, (3beta)-, Stigmasterol, Campesterol, and beta-Sitosterol also showed H-bond interactions with binding affinity -8.1, -7.5, -6.9, and -6.9 respectively while Ursa-9(11),12-dien-3-ol and 11-Oxours-12-en-3-yl acetate didn’t show any H-bond interactions. This demonstrates *F. vasta’s* ability to inhibit α-glucosidase and α-amylase. Figure 5, Figure 6 and Appendix A, and Table 8 illustrate the docking results of the compounds with both receptors. The docking study was validated by superimposing the co-crystallized ligand (Acarbose) with extracted Acarbose and redocked to α-amylase (PDB: 1b2y) crystal structure and a low RMSD of 1.525 Å was observed. Similarly, an RMSD of 1.234 Å was observed with α-glucosidase (pdb id: 3TOP). The co-crystallized Acarbose (Yellow) and extracted Acarbose (Red) are shown in Appendix A.

#### 2.3.2. ADMET Analysis

The most promising docked compounds were further investigated by the SwissADME online tool, which provides information on pharmacokinetics, drug-likeness, and physicochemical properties. According to Lipinski’s rule, all compounds violated the rule of lipophilicity while Olean-12-en-3-ol, acetate, (3beta)- and Stigmasterol violated the rule of molar refractivity. When a drug fails to meet two or more criteria, it is classified as a non-orally available drug. However, all of the compounds had one violation, indicating that they are orally available or bioavailable drugs. According to Lipinski’s criteria, all of the compounds exhibited orally active likeness characteristics. All of the compounds had no Blood-brain barrier penetration and lowed GI absorption. However, this study does not determine whether a compound has a specific biological effect. Table 9 presents the properties of the best seven best-docked compounds, such as molecular weight, no. of hydrogen bond acceptor and doner, no. of rotateable bond, Lipinski rule, and lipophilicity, while Figure 7 presents the bioavailability radar of best-docked compounds. Appendix A tabulates the pharmacokinetic properties of best-docked compounds. Olean-12-en-3-ol, acetate, (3beta)-, and 11-Oxours-12-en-3-yl acetate were predicted to have carcinogenic and immunotoxic properties in ProTox-II. In this investigation, all compounds were predicted to have minimal toxicity, except beta-Amyrin, which was predicted to be non-toxic. Appendix A presents different compounds along with their predicted toxicity class and LD50 value.

## 3. Discussion

The phytochemicals profiling showed that the extract of *F. vasta* is a good source of carbohydrates, alkaloids, steroids, glycosides, flavonoids, phenols, tannins, and saponins. The antioxidant, antimicrobial, antifungal, antidiabetic, anti-inflammatory, anti-cancer, and antihypertensive effects are known to exist in these recognized phytochemical classes [58]. Some classes such as alkaloids have anti-viral, antifungal, antitumor, and antimicrobial activity [59]; flavonoids, phenols, and tannins have antioxidant and anti-cancer potential [60]; and saponins have antidiabetic, antibacterial, anti-inflammatory, and anti-cancer properties [61]. These phytochemicals in *F. vasta* extract may play a part in its therapeutic benefits. *Ficus* genus is medically important due to the presence of high value of total bioactive contents (TPC and TFC) [62]. Previously, some flavonoid compounds were identified in leaves of *F. vasta* [25]. The high value of total phenolic and flavonoid contents in plant extract correlates with the strong antioxidant activities [63].

To perform in-depth phytochemical identification, the ethanolic extract was evaluated by gas chromatography-mass spectrometry. The GCMS chromatogram of *F. vasta* ethanolic extract showed 28 peaks of compounds. Major compounds of GCMS consisting of Stigmasterol, 22,23-dihydro, 11-Oxours-12-en-3-yl acetate, Phytol, n-Hexadecanoic acid, Quinic acid, 11 14 17-eicosatrienoic acid methyl ester, 4-((1E)-3-Hydroxy-1-propenyl)-2-methoxyphenol, and 2,3-Dihydrobenzofuran. In GCMS, most of the compounds related to the classes like fatty acids, steroids, vitamins, and esters. The compounds identified by GCMS were reported to possess different biological activities. On this basis, we decided to examine antioxidant, antibacterial, antifungal, anti-viral, cytotoxicity, and antidiabetic activities. 

According to the literature, the antioxidant activity of the ethanolic extract in aerial parts of *F. vasta* has not been reported, while the methanol extract of *F. vasta* leaves showed the antioxidant activity [25]. The results of the DPPH, ABTS, FRAP, and CUPRAC activities showed that there is a direct link of antioxidant activities and the polyphenol contents (i.e., high value of phenolic and flavonoid contents correlates with the strong antioxidant activity) [63]. 

The α-amylase initiates carbohydrate digestion by hydrolyzing 1,4-glycosidic bonds in polysaccharides to disaccharides, which is followed by α-glucosidase catalyzing the disaccharides to monosaccharides, resulting in postprandial hyperglycemia [64,65]. As a result, α-amylase and α-glucosidase inhibitors can control hyperglycemia by delaying carbohydrate digestion and lowering postprandial plasma glucose levels [66]. There is no data in the literature about the in vitro antidiabetic studies of *F. vasta*. Our results showed that ethanolic extract of *Ficus vasta* exhibits excellent α-glucosidase and α-amylase inhibitory effects. Such significant inhibition of α-amylase and α-glucosidase may be due to the presence of some phytocompounds which were identified by GCMS profiling, such as Ursa-9(11),12-dien-3-ol and beta-Amyrin which showed significant binding affinities with these enzymes, and due to presence of some other compounds in the extract. This suggests the potential of *F. vasta* for the management of diabeties.

The antibacterial ability of *F. vasta* was tested against five gram-positive strains and two gram-negative strains. Most of the bacteria employed in the assay showed a significant zone of inhibition (>9 mm) in the results. The antifungal ability of *F. vasta* was tested against three strains, including *Aspergillus niger*, *Fusarium avenaceum*, and *Fusarium brachygibbosum*. Most of the fungi employed in the test showed a significant zone of inhibition in the results. The anti-viral potential against three strains (i.e., avian infectious bronchitis virus (IBV), Influenza virus (H9), and Newcastle disease virus (NDV)) was carried out using a haemagglution test method. The ethanolic extract of *F. vasta* was effective and strongly active against the viral strains. The presence of phytochemical components such as alkaloids, phenolic and flavonoid compounds could be responsible for the antimicrobial activities [67]. Tentative identification of the ethanolic extract by GCMS revealed many compounds with antibacterial, antifungal, and antiviral activities, namely Methyl (Z)-5,11,14,17-eicosatetraenoate [4,8], Octadecanoic acid [45], 4-((1E)-3-Hydroxy-1-propenyl)-2-methoxyphenol [34], Nonanoic acid [31], Catechol [27], and Stigmasterol, 22,23-dihydro [54] etc. Methanolic extract obtained from leaves of *F. vasta* previously tested against *E. coli*, *Pseudomonas aeruginosa*, *Streptococcus pneumonia*, *Salmonella typhimurium*, and *Staphylococcus epidermidis* [25,68].

To the best of our knowledge, there is no information in the literature about the ethanolic extract of *F. vasta* thrombolytic activity. Plants containing flavonoids and polyphenols have been reported to possess thrombolytic activity and hence attracted researchers to explore more for better and safer drugs of plant origin [69]. The thrombolytic activity of *F. vasta* was significant. Microbial plasminogen activators such as staphylokinase and streptokinase function as co-factoring molecules to promote development [70].

Hemolysis is the breakdown or disruption of the integrity of the RBC membrane, which causes hemoglobin to be released from red blood cells [71]. Some traditional plants sometimes become harmful if used over an extended period. There are chemical elements in many plants that may have an anti-hemolytic or hemolytic effect on human erythrocytes. Plant extracts can disrupt red blood cell membranes, leading to serious side effects such hemolytic anemia [72]. Therefore, it is important to evaluate the possible hemolytic activity of a number of the frequently used herbs. When there is more than 30% hemolysis, the plant extracts are thought to be harmful to erythrocytes [73]. Table 7 displays the hemolytic activity of aerial parts of the entire *F. vasta* plant. Plant hemolysis was measured as a percentage of hemolysis. The results showed that extract had the hemolytic activity (6.39 ± 0.45%). Since the hemolysis activity of extract is less than 30%, they are all nontoxic and safe for human utilization. This is the first time that the *F. vasta* plant’s hemolytic activity has been reported.

MTT activity was used to evaluate the effect of *F. vasta* extract on HepG2 cell growth. Treatment with *F. vasta* extract for 48 h inhibited the growth of cells in a dose-dependent manner. Our results found that *F. vasta* is rich in antioxidants which can act as scavengers for free radicals created by the body’s cells as a result of a variety of metabolic reactions and external influences. These antioxidants are useful to cells since they help to prevent the growth of cancer [12].

In recent years, computer-based modeling techniques have been increasingly used to predict the interactions between small molecules and their biological targets. This approach can provide useful insights into the molecular basis of the biological activity of natural products and the possible mechanisms of action and binding modes of active compounds [74]. To understand the ability of the compounds to inhibit enzymes and to find a correlation between the in vitro enzyme inhibition results, all compounds from the GCMS analysis of ethanolic extract were docked against α-amylase and α-glucosidase enzymes, along with acarbose. According to the validation criteria, RMSD values less than 2.0 Å demonstrate that the docking protocol is capable of accurately predicting the co-crystallized ligand’s binding orientation [75]. The hydrogen bond and other hydrophobic interactions, such as alkyl and pi-alkyl, plays an important protein-ligand interactions as well as ensuring the stable binding of ligands with proteins [76].

For further investigation, the best docked compounds were studied using online tool SwissADME which gave information about their pharmacokinetics, drug likeness, and physiochemical properties [77]. According to Lipinski, a chemical can exhibit drug-like behaviour if it meets all of the following criteria: (i) Molecular weight (<500); (ii) Hydrogen bond donor (≤5); (iii) hydrogen bond acceptor (less than and equal to 10); (iv) Lipophilicity (Log Po/w, <5); and (v) molar refractivity 40 to 130. Those compounds which followed the Lipinski rule were considered potential therapeutic candidates [78]. When a drug fails to meet two or more of Lipinski’s criteria, it is classified as non-orally available drug. However, all of the compounds had 1 violation, indicating that they are bioavailable or orally available drugs. According to Lipinski’s criteria, all of the best docked compounds exhibited orally active likeness characteristics. Compounds with lower lipophilicity, molecular weight, and hydrogen bond capacity are said to have good absorption, high permeability, and bioavailability [79,80]. All of the compounds had no blood-brain barrier penetration and low GI absorption. However, this study does not determine whether a compound has a specific biological effect. The colors in the bioavailability radar of best docked compounds represent the best physico-chemical space indicator for oral bioavailability, taking into account factors such as lipophilicity, saturation, size, flexibility, polarity, and solubility. LogP is a measure of lipophilicity, can range from −0.7 to +5.0, molecular weight can range from 150 to 500 (g/mol). The TPSA, which is a measure of the size and polarity of a molecule, ranges from 20 to 130 A^2^. The logS (ESOL) insolubility, which measures the solubility of a molecule, ranges from 0 to 6. The number of rotatable bonds for a molecule is between 0 to 9, with an unsaturation fraction between 0.25 and 1.0. This indicates that the carbon atom fraction in sp3 hybridization cannot be less than 0.25 [81]. ProTox-II is a program that predicts toxicity based on the similarity of chemical structures and then compares them to other chemicals with known toxicities [82]. In silico toxicity analysis of seven phytocompounds showed that they all have low toxicity potential. However, beta-Amyrin was predicted to be non-toxic.

Overall, Ursa-9(11),12-dien-3-ol, Olean-12-en-3-ol, acetate, (3beta)-, beta-Amyrin, 11-Oxours-12-en-3-yl acetate, Campesterol, beta-Sitosterol, and Stigmasterol may have some potentials as inhibitors of important proteins (α-glucosidase and α-amylase) and might have contributed either singly or in synergy to the antidiabetic properties of *F. vasta*. However, further research is needed to explore the pharmacokinetic properties of these compounds and whether or not they are effective in both in vitro and in vivo models.

## 4. Materials and Methods

### 4.1. Materials and Reagents

Chemicals used in the study were of analytical grade purchased from Sigma Aldrich, Louis, MO 63103, USA. All of the standards were also purchased from Sigma.

### 4.2. Collection and Extraction of Plant

The aerial parts of *F. vasta* were collected in September 2020 from Jazan province, Fayfa Mountains, Saudi Arabia. A botanist identified the plant at the Science College, King Saud University, Saudi Arabia. A Voucher specimen No. 24557 was submitted to the herbarium of the Science College, King Saud University. The plant was air-dried and then subjected to crushing and grinding to form a coarse powder. Air-dried plant was macerated with 80% ethanol for 15 days. After maceration, it was filtered and then concentrated (40 °C) using a rotary evaporator (Heidolph, Germany), yielding a dark brown residue and further air dried to yield 98.5 g extract.

### 4.3. Phytochemical Analysis

#### 4.3.1. Preliminary Phytochemical Profiling

The presence of secondary and primary metabolites such as alkaloids, steroids, terpenoids, resins, carbohydrates, amino acids, phenols, lipids, proteins, saponins, glycosides, tannins, and flavonoids was determined by qualitative phytochemical analysis of *F. vasta* extract [83].

#### 4.3.2. Total bioactive contents (TPC and TFC)

##### TPC

The Folin-Ciocalteu method was used to determine the total phenolic content (TPC) as previously reported in the literature [84]. Different concentrations of Gallic acid (0.05-0.5 mg/mL) were used as a standard to establish a standard calibration curve (Appendix A). The sample solution was prepared as 0.5 mg/mL, from which an aliquot of 0.1 mL was taken in a test tube, followed by adding Folin-Ciocalteu’s reagent (0.1 mL). Then, 10% Na_2_CO_3_ (2.8 mL) was combined with the resultant solution, and the solution was placed for 30 min in the dark. Absorbance was read at 765nm using a BioTek Synergy HT microplate reader. TPC was expressed in milligrams of Gallic acid equivalent per gram of dry extract.

##### TFC

The aluminum chloride colorimetric method was used to estimate the content of flavonoids [84]. In 96 % ethanol, standard quercetin solutions of 30, 40, 50, 60, 70, 80, 90, 100 µg/mL were prepared (Appendix A). Firstly, the 1 mg/mL solution of extract and then 50 µL extract and standard solutions were added in 10 µL of aluminium chloride (10 %) solution, and then 150 µL of ethanol (96%) was added. At last, in 96 well plates, 10 µL of sodium acetate (1M) was added to the mixture. Ethanol was used as a blank, and then all reagents were mixed, and the mixture was then incubated for 40 min at room temperature in the dark. Finally, the absorbance was measured at 415 nm using a BioTek Synergy HT microplate reader. TFC was expressed in milligrams of quercetin equivalents per gram of dry extract.

#### 4.3.3. Gass Chromatography-Mass Spectroscopy Analysis

The phytometabolites of ethanolic extract of *F. vasta* were determined by GCMS analysis using Agilent, 6890 series, and Hewlett Packard, 5973 mass selective detector. An HP-5MS column with a length of 30m, a diameter of 250 µL, and a film thickness of 0.25 µL were used to achieve the best possible separation. The volume of 1.0 µL of the extract was diluted with the appropriate solvent and injected at 250 °C in a splitless mode. Helium gas as a carrier was used at a constant flow rate of 1.02 mL/min., the temperature was increased gradually, starting at 50–150 °C and increasing by 3 °C per min, with a 10 min holding time at each temperature. The final temperature was set to 300 °C at 10 °C/min. The components were identified using their retention indices, and the mass spectrum was interpreted using the National Institute of Standards and Technology database (NIST) [85].

### 4.4. Biological Activities

#### 4.4.1. Antioxidant Activities

The antioxidant activities of *F. vasta* included determining the free radical scavenging and reducing power activities. Ascorbic acid was used as a standard for all antioxidant activities. The DPPH, ABTS, FRAP, and CUPRAC activities were performed according to the literature with minor modifications [86].

##### DPPH Assay

A solution of 0.1 mM DPPH was prepared, and 90 µL of DPPH solution was added to a 96-well plate, followed by 10 µL of the substance being tested. The mixture is then incubated for 30 min, after which the absorbance is read at 517 nm using a BioTek Synergy HT microplate reader.

##### ABTS Assay

ABTS activity was carried out by the modified method in which 100 µL of sample solution was mixed with 200 µL of prepared ABTS solution. This mixture was incubated for 30 min at room temperature. The absorbance was recorded at 417 nm using a BioTek Synergy HT microplate reader.

##### FRAP Assay

For this activity, 100 µL of *F. vasta* extract (1mg/mL) was mixed with 2 mL of FRAP reagent and mixture was incubated for 30 min. The absorbance was measured at 593 nm using a BioTek Synergy HT microplate reader. Similarly, the blank sample was prepared without extract.

##### CUPRAC Assay

For this activity, 100 µL solution of *F. vasta* extract was prepared, and added into 7.5 mM (200 µL) neocuprine, 10 mM (200 µL) CuCl_2,_ and 1M (200 µL) ammonium acetate buffer (pH 7.5) reaction mixture. This mixture was incubated for 30 min then absorbance was recorded at 450 nm using a BioTek Synergy HT microplate reader.

#### 4.4.2. Enzyme Inhibition Activities

##### α-Glucosidase Inhibition Assay

The method described in previous literature was followed to determine the α-glucosidase inhibitory activity [87]. In this assay acarbose was used as a standard or positive control and methanol as a negative control. The volume of 10 µL of a plant extract, 70 µL of a 0.1 molar phosphate buffer (pH 6.8), and 10 µL of α-glucosidase was added to the wells of a 96 well plate and incubated for 15 min at 30 °C. In the end, 10 µL p-Nitrophenyl-α-D-glucopyranoside solution was added for the next 30 min and absorbance was read at 405 nm using a BioTek Synergy HT microplate reader.

##### α-Amylase Inhibition Assay

According to the previously reported method [88], the starch iodine test method was used to evaluate the inhibitory activity of α-amylase. 25 µL of 0.02 molar sodium phosphate buffer containing 6 millimolar NaCl, 20 µL of soluble starch (concentration 1% *w*/*v*), and 20 µL of plant extract/acarbose was incubated at 37 °C for 5 min. Acarbose was used as a standard. Then 15 µL of amylase solution were placed into each well and incubated at 37 °C for the next 10 min. After that, 20 µL of 1 molar was added to stop the reaction. Then, 100 µL of iodine reagent was added. The color change was observed, and absorbance was measured at 620 nm using a BioTek Synergy HT microplate reader. The color of the mixture changes based on the presence of starch. If there is an active inhibitor, the color will be dark blue. If there is no inhibitor, the color will be yellow. If the starch is partially degraded, the color will be brownish, indicating that the inhibitor is partially active.

#### 4.4.3. Antibacterial Activity

##### Strains of Bacteria

Antibacterial activity was performed against the five positive strains, including Bacillus subtilis, Bacillus pumilus, Staphylococcus epidermidis, Staphylococcus aureus, Micrococcus luteus, and two negative strains, including Escherichia coli, and Bordetella bronchiseptica. These bacterial strains were provided by the Microbiology lab of the Islamia University of Bahawalpur.

##### Agar Well Diffusion

The culture of bacteria was streaked on Mueller Hinton agar plates and placed in an incubator for 24 h at 37 °C. After 24 h, the colony was picked and inoculated into the saline solution and vortexed. In the end, turbidity was set to 0.5 McFarland standards. The plant extracts were prepared in DMSO at four concentrations 25, 50, 75, and 100 mg/mL. Mueller Hinton agar plates were inoculated with various strains of bacteria. The organisms were spread evenly over the surface of the agar, and four wells (6 mm in diameter) were punched into each plate. 100 µL of sample solution was added to each well, and the plates were incubated at 37 °C for 24 h. The diameter millimeter of a zone of inhibition was measured and ceftriaxone was used as standard in the activity [89].

#### 4.4.4. Antifungal Activity

##### Fungal Strains

Antifungal activity was performed against the three strains, including *Aspergillus niger*, *Fusarium avenaceum*, and *Fusarium brachygibbosum*. These fungal strains were provided by the Microbiology lab of the Islamia University of Bahawalpur.

##### Agar Tube Dilution

Antifungal activity was performed according to the previous method with little modifications [90]. The sample solutions were prepared in DMSO at 20 and 40 mg/mL concentrations. Agar was prepared, and the sample solution was added to melted agar. Then tubes were solidified at room temperature in a slanting position. After solidification, each slant was inoculated by 4 mm diameter piece of fungal strain. Terbinafine was used as standard and DMSO as the negative control in the activity. Tubes were incubated for 3 days at 27–29 °C. The percentage inhibition was measured with the following formula:(1)$$\mathrm{Percentage}\;\mathrm{inhibition}=\frac{\mathrm{Linear}\;\mathrm{growth}\;\mathrm{in}\;\mathrm{test}\;\mathrm{tubes}\;\left(\mathrm{mm}\right)}{\mathrm{Linear}\;\mathrm{growth}\;\mathrm{in}\;\mathrm{control}\;\left(\mathrm{mm}\right)}\times 100$$

#### 4.4.5. Anti-viral Activity

##### Viral Strains

Anti-viral activity was performed against the three strains including Avian Infectious Bronchitis Virus (IBV, H120 strain from IZO S.U.R.L 99/A-25124), Newcastle disease virus (NDV, Lasoota strain), and Influenza virus (H9. H9N2 strain) from Brescia, Italy.

##### Inoculation of Viruses in Chicken Embryonated Eggs

The reported procedure was performed for anti-viral activity in ethanolic extract of *F. vasta* [91]. Chicken eggs are used for preliminary growth of the viruses and formation of novel vaccines. Eggs are considered as one of the most used media due to excellent growth of viruses. The chicken eggs are easy to collect, easy to handle, aseptic conditions, cheep, and require little space for storage making them the best resource of viral inoculations studies. The viruses accumulate in the chorioallantoic membrane fluid during incubation period of the eggs and keep replicating. Viral strains were cultured in the 7 to 11 days old embryonated poultry eggs. Pathogen-free eggs were collected from Government Poultry Farm, Model Town a, Bahawalpur. Ethyl alcohol was used to sterilize the eggs, and a sterile needle was used to make holes in the egg. Viral strains were injected through the chorioallantoic route into the embryonated eggs with the help of a syringe (5 cc). After inoculation, the hole was closed with melted wax after the inoculation. The eggs were incubated at 37 °C for 48–72 h. Melted wax was then used to close the hole. The incubation period for the inoculated eggs was 48 to 72 h at 37 °C. Titer viruses were then assessed after collecting the allantoic fluid with the syringe in the Eppendorf at 4 °C for further processing. A 96-well microtiters round bottom plate was used for performing the Haemagglutination test.

##### Heamagglutination (HA) Test

The first step of the Haemagglutination test is to prepare 1% red blood cells (R.B.Cs.) Alsever solution was poured into a test tube, and fresh chicken blood (5 mL) was added. Blood (5 mL) was centrifuged for 5 min at 4000 rpm, and the supernatant was removed. For better results and more purification, the process was repeated three times. In the Eppendorf tubes, 10 µL packed R.B.Cs were mixed with 1 mL phosphate buffered saline solution at pH 7.4 to prepare a 1% R.B.Cs solution. To avoid precipitation, Eppendorf tubes were shaken gently. In each microtiter plate well, PBS (50 µL) was added. The samples (50 µL) were added in the first column. The diluted upto the 11th and 12th well was left as negative control only containing PBS. After that, 1% RBCs solution (50 µL) was added to all 12 wells, and the plate was incubated for 2 to 3 h at 37 °C. The uniform red color indicated positive results; however, red dots at the bottom of the well indicate negative results. 56 Haemagglutination Titer had the highest dilution number showing positive results. The test was used for testing the titer of Avian Infectious Bronchitis Virus (IBV), Newcastle disease virus (NDV), and Influenza virus (H9).

#### 4.4.6. Thrombolytic Activity

The blood samples were taken in the incubated Eppendorf tubes for 45 min at 37 °C. Serum was removed from the blood sample without damaging the blood clot, and the Eppendorf tube was weighed again. In each pre-weighed Eppendorf tube with a blood clot, 1 mg/mL (100 µL) of plant extract was added. Streptokinase was used as a standard and negative non-thrombolytic control. Eppendorf tubes containing blood samples and plant fractions were incubated at 37 °C for 90 min. After 90 min, the thrombolytic activity was observed in these Eppendorf tubes. The fluid released in the Eppendorf tubes was removed and weighed again. The weight difference of Eppendorf tubes showed the antithrombotic activity of the extract or fractions against streptokinase [92]:(2)$$Percentage\;of\;clot\;lysis=\left(\frac{Wr}{Wc}\right)\times 100$$
where: *Wr* = released clot weight; *Wc* = clot weight

#### 4.4.7. Hemolytic Activity

The hemolytic activity was performed according to the literature with some modifications [93]. A hemolytic activity on human erythrocytes was used to determine the preliminary toxicity of phytochemical compounds from plant extract. To prepare the erythrocyte suspension, blood (type O blood from Civil Hospital, Bahawalpur) was used. The volume of 50 µL of the erythrocyte suspension (pH 7.4) and 100 µL of the extract were mixed and incubated for 1 h at 37 °C. After that, 850 µL of phosphate-buffered saline (pH 7.4) was added in already prepared mixture, then centrifuged for 3 min at 3000 rpm. The supernatant was obtained, and the absorbance at 540 nm of the hemolysis was measured by using a BioTek Synergy HT microplate reader. A total of 0.1% Triton X-100 was used as a positive control while phosphate-buffered saline was used as a negative control:(3)$$\mathrm{Hemolysis}\;\mathrm{percentage}\;=\frac{\mathrm{Absorbance}\;\mathrm{of}\;\mathrm{sample}-\mathrm{Absorbance}\;\mathrm{of}\;\mathrm{negative}\;\mathrm{control}}{\mathrm{Absorbance}\;\mathrm{of}\;\mathrm{positive}\;\mathrm{control}}\times 100$$

#### 4.4.8. Cytotoxicity

The human HepG2 liver cancer cells were grown in the Dulbecco’s Modified Eagle’s Medium (DMEM), with added 10% fetal bovine serum (FBS), 100 IU/mL penicillin, and 100 μg/mL streptomycin in a 75 cm^2^ flask and maintained at 37 °C with 5% CO_2_ incubator. Cells were treated with extracts dissolved in DMSO (0.05% concentration).

##### Determination of Cell Viability

Cell viability was determined by MTT activity as described in the literature with some modifications [94]. Firstly, HepG2 cells were treated with different concentrations from 10 to 100 ug/mL of extracts for 48 h. Following treatment, each well was introduced with a 10 µL MTT reagent and further incubated for 4 h. Then 150 μL of DMSO was subsequently introduced to dissolve formazan crystals, and absorbance was read at 490 nm. The % age of cell viability was calculated:(4)$$\mathrm{Cell}\;\mathrm{viability}\;\%\;=\frac{\mathrm{Absorbance}\;\mathrm{of}\;\mathrm{sample}-\mathrm{Absorbance}\;\mathrm{of}\;\mathrm{blank}}{\mathrm{Absorbance}\;\mathrm{of}\;\mathrm{control}-\mathrm{Absorbance}\;\mathrm{of}\;\mathrm{blank}}\times 100$$

### 4.5. In Silico Activities

#### 4.5.1. Molecular Dockings

The interaction between the bioactive compounds discovered in *F. vasta* ethanolic extract, α-glucosidase (PDB: 5zcb), and α-amylase (PDB: 4w93) was assessed using PyRx software. The 3D shapes of proteins were taken from the RCSB Protein Data Bank (https://www.rcsb.org/ accessed on 30 May 2022). First, the proteins were prepared for docking using the Discovery studio by removing water and ligands, inserting polar hydrogen atoms, and saving them in PDB format. Then download the ligand in SDF format from PubChem (https://pubchem.ncbi.nlm.nih.gov accessed on 30 May 2022). The Open Babel software converted the ligands to PDB files. The prepared and optimized ligands were docked blindly in the protein’s grid box to allow them to find any suitable binding location [95]. The docking studies were validated by superimposing the co-crystallized ligand (Acarbose) with extracted Acarbose from crystal structure and redocked to α-amylase and α-glucosidase crystal structures. The Ligplot software was used to visualize 2D structures of ligand-protein interactions [96].

#### 4.5.2. ADMET Analysis

The ADME characteristics of the best docked bioactive compounds were assessed using the online SwissADME (http://www.swissadme.ch/ accessed on 12 June 2022) [77]. The best-docked compound’s toxicity was checked using the online tool PROTOX II (https://tox-new.charite.de/, accessed on 12 June 2022) [82].

### 4.6. Statistical Analysis

The findings were all presented as mean standard deviation (mean ± SD). The information obtained through quantitative analysis IBMSPSS (v20, Chicago, IL, USA) was used to perform a one-way analysis of variance (ANOVA) followed by post-hoc test on the phytochemicals. Significant values were defined as p values less than 0.05.

## 5. Conclusions

The results of the present study suggest that ethanolic extract of *Ficus vasta* Forssk. has identified bioactive phytochemicals responsible for therapeutic and pharmacological activities. *F. vasta* showed the good TPC, TFC, and antioxidant potential and had a good antidiabetic potential for α-amylase and α-glucosidase inhibition. The extract exhibited the good antibacterial, antifungal, and anti-viral potentials against the strains. *F. vasta* extract inhibited the growth of HepG2 cells in a dose-dependent manner. The GCMS analysis of ethanol extract indicated the tentative presence of important phytocompounds, which justified their biological activities. Hence, it is concluded that in vitro analysis of *F. vasta* showed the plant’s medicinal potential with respect to antioxidant, antidiabetic, thrombolytic, enzyme inhibition, cytotoxicity, anti-viral, antifungal, and antibacterial potential. The in silico molecular docking studies further explained the enzyme inhibition activity. Based on in vitro and in silico docking studies, further investigation is required to evaluate its toxicity profile and clinical studies. Conclusively, the findings of this research could help researchers who are struggling continuously for the development of novel and effective drugs from natural products. The observed phytochemical and biological potential of this plant indicated that it might be valuable for further isolation of bioactive compounds.

## Figures and Tables

**Figure 1 antibiotics-11-01155-f001:**
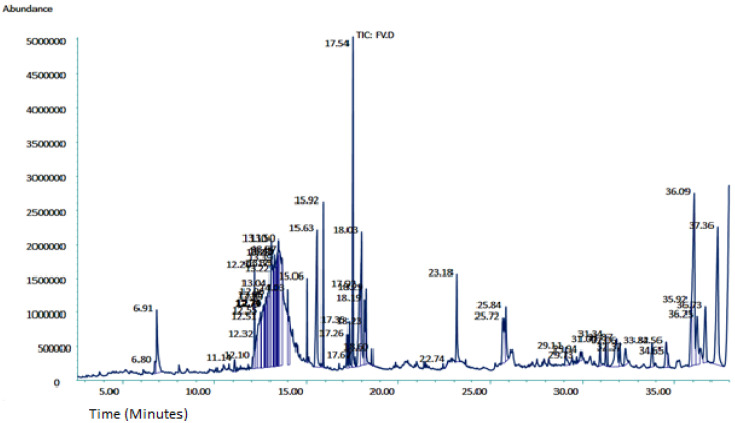
GCMS chromatogram of *F. vasta* ethanolic extract.

**Figure 2 antibiotics-11-01155-f002:**
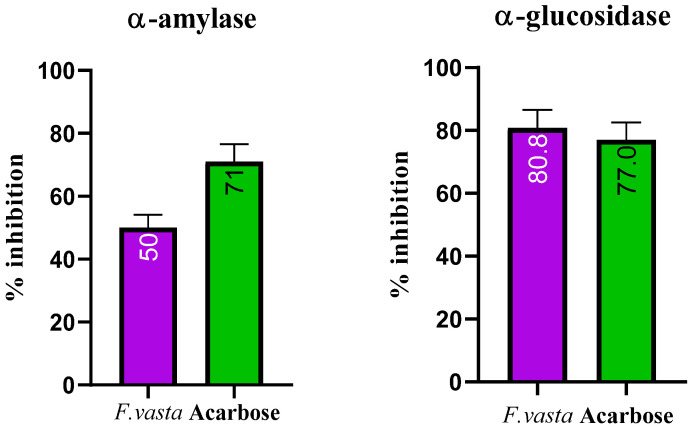
% Inhibition of α-amylase and α-glucosidase in the extract of *F. vasta*.

**Figure 3 antibiotics-11-01155-f003:**
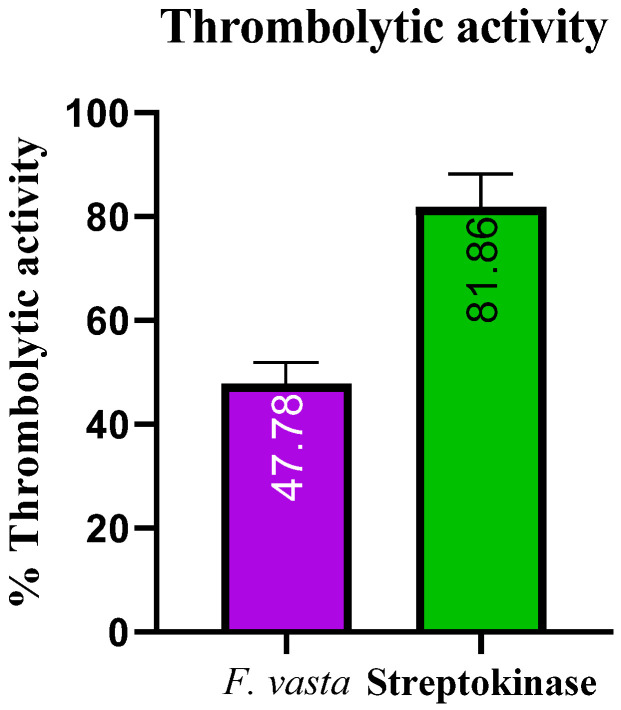
The thrombolytic activity of *F. vasta*.

**Figure 4 antibiotics-11-01155-f004:**
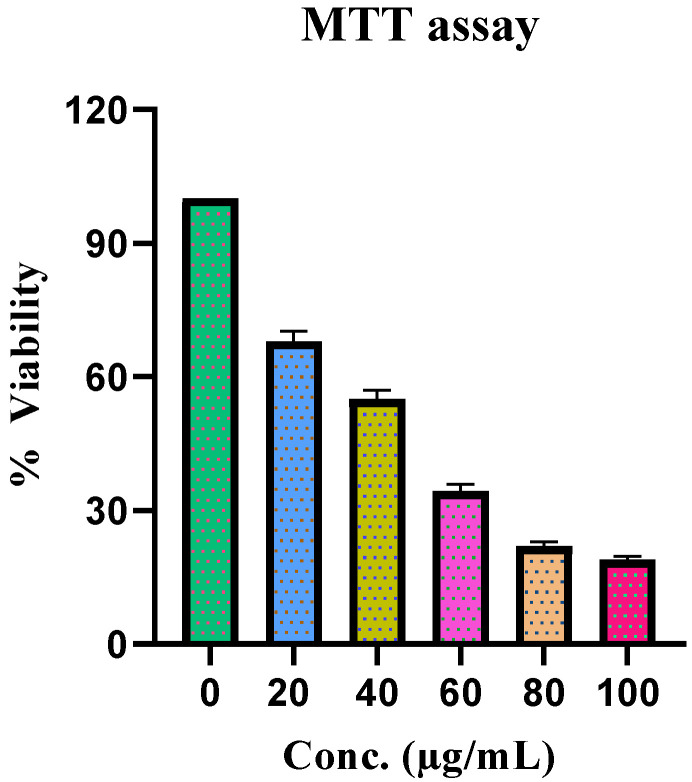
Cytotoxicity activity of *F. vasta*.

**Figure 5 antibiotics-11-01155-f005:**
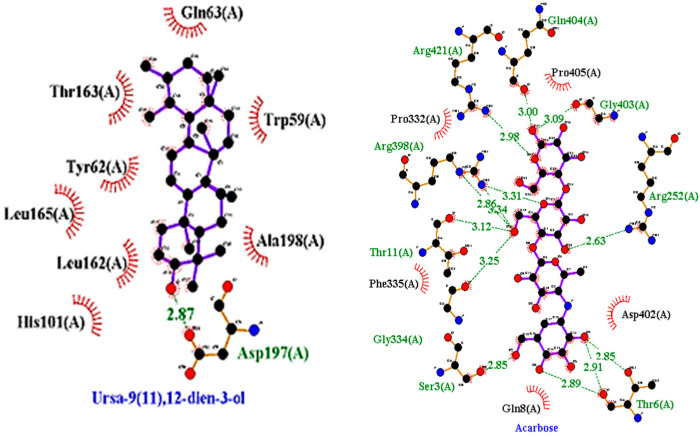
2D structures of best-docked compound and Acarbose against α-amylase.

**Figure 6 antibiotics-11-01155-f006:**
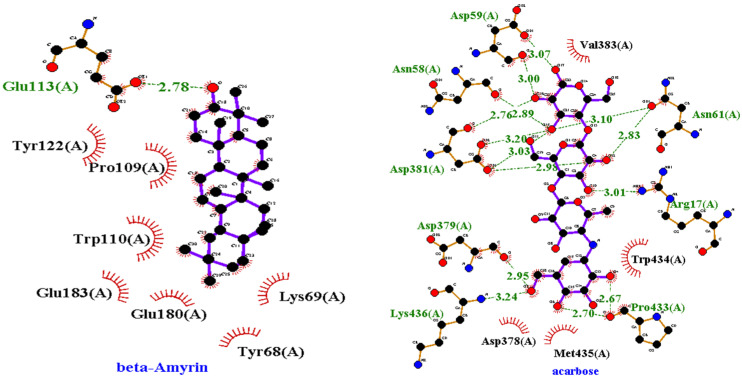
2D structures of best-docked compound and Acarbose against α-glucosidase. The green dotted line showed the hydrogen bond, while the red showed the hydrophobic interactions.

**Figure 7 antibiotics-11-01155-f007:**
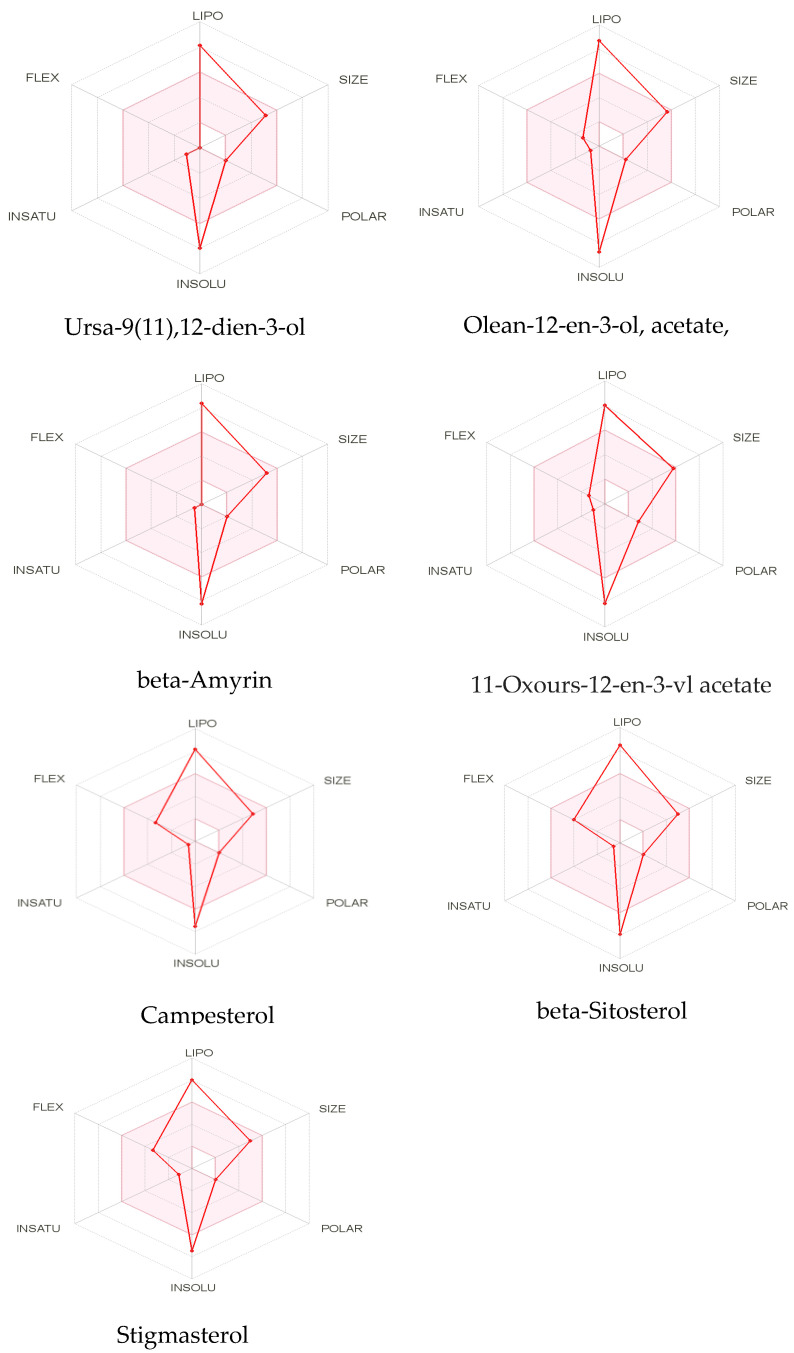
The bioavailability radar of best-docked compounds. The pink area represents oral bioavailability, considering factors such as polarity, lipophilicity, saturation, size, flexibility, and solubility.

**Table 1 antibiotics-11-01155-t001:** Preliminary phytochemical profiling of *F. vasta*.

Sr. No.	Metabolites	Tests	Results
1	Carbohydrates	MolischTest	+
		Iodine Test	+
2	Proteins	Biurette Test	−
3	Lipids	Saponification Test	+
4	Flavonoids	Reaction with NaOH	+
5	Tannins	Lead Acetate Test	+
6	Saponins	Frothing	+
7	Amino acids	Ninhydrin Test	−
8	Steroids/Terpenes	SalkowaskiTest	+
9	Glycosides	Erdmann Test	+
10	Alkaloids	Hager Test	+
		Wagner Test	+
		Mayer Test	+
11	Phenols	Ferric chloride Test	+
12	Resins	Acetic Anhydride Test	−

“+” Present and “−” Absent.

**Table 2 antibiotics-11-01155-t002:** Total bioactive contents and antioxidant activities of *F. vasta* extract.

Sample	TPC mg GAE/g	TFC mg QE/g	Antioxidant Activities (mg/mL) IC_50_			
			DPPH	FRAP	ABTS	CUPRAC
*F. vasta*	89.47 ± 3.21	129.2 ± 4.14	1.75 ± 0.08	1.91 ± 0.11	1.63 ± 0.06	1.51 ± 0.04

All tests were performed in triplicates and results are expressed as mean ± S.D.

**Table 3 antibiotics-11-01155-t003:** Phytocompounds identified in ethanolic extract of *F. vasta* by GCMS.

Sr No.	RT (minutes)	Compounds Identified	M. Formula	Mol. Wt (g/mol)	Chemical Class	Area (%)	Reported Activities of Compounds
1	6.80	Catechol	C_6_H_6_O_2_	110.11	Benzenediol	0.25	Antioxidant [26], antibacterial, and antifungal [27]
2	6.91	2,3-Dihydrobenzofuran	C_8_H_8_O	120.15	1-Benzofurans	2.17	Antileishmania [28]
3	12.20	4,4,5,8-Tetramethylchroman-2-ol	C_13_H_18_O_2_	206.28	Vitamin E analog	1.39	Anti-inflammatory [29]
4	12.74	Levoglucosan	C_6_H_10_O_5_	162.14	Carbohydrate	0.55	Antibacterial [30]
5	12.76	Nonanoic acid	C_9_H_18_O_2_	158.24	Ester	0.58	Antimicrobial [31], antifungal [32]
6	12.79	Phloroglucinol	C_6_H_6_O_3_	126.11	Benzene triol	0.40	Oxidative stress [33]
7	13.10	4-((1E)-3-Hydroxy-1-propenyl)-2-methoxyphenol	C_10_H_12_O_3_	180.2005	Organic compound	3.71	Anti-inflammatory, antimicrobial, and antioxidant [34]
8	13.50	Quinic acid	C_7_H_12_O_6_	192.17	Cyclitol/Cyclohexane carboxylic acid	4.07	Anti-carcinogenic and antioxidant [35]
9	15.63	n-Hexadecanoic acid	C_16_H_32_O_2_	256.42	Fatty acid	4.27	Antioxidant, and nematicide [36]
10	15.92	Hexadecanoicacid, ethyl ester	C_18_H_36_O_2_	284.5	Fatty acid ester	1.82	Antioxidant, and nematicide [37]
11	17.35	9,12,15-Octadecatrienoic acid, methyl ester, (Z,Z,Z)-	C_19_H_32_O_2_	292.5	Fatty acid ester	0.72	Anti-inflammatory, anti-cancer, hepatoprotective, [38]
12	17.54	Phytol	C_20_H_40_O	296.5	Diterpenoid	5.13	Cytotoxic, antioxidant, and antimicrobial [39,40]
13	17.67	16-Methylheptadecanoic acid methyl ester	C_19_H_38_O_2_	298.5	Ester of isostearic acid	0.13	Anti-cancer [41]
14	17.92	9 12-Octadecadienoic acid (z z)- methyl ester	C_19_H_34_O_2_	294.4	Fatty acid	2.80	Hepatoprotective [42]
15	18.03	11 14 17-eicosatrienoic acid methyl ester	C_21_H_36_O_2_	320.5	Fatty acid methyl ester	3.76	Anti-inflammatory, anti-arthritic [43]
16	18.19	Linoleic acid ethyl ester	C_20_H_36_O_2_	308.5	Linoleic acid	0.99	Anti-acne [44]
17	18.23	Octadecanoic acid	C_18_H_36_O_2_	284.5	Fatty acid (stearic acid)	0.43	Antioxidant, antimicrobial [45]
18	22.74	12-Oleanene-3-yl acetate (3.α.)-	C_32_H_52_O_2_	468.8	Triterpenoid	0.01	Antioxidant and cytotoxic [46]
19	23.18	Hexadecanoicacid, 2-hydroxy-1-(hydroxymethyl)ethyl ester	C_19_H_38_O_4_	330.5	Fatty acid	1.77	Antioxidant, and anti-inflammatory [47]
20	25.84	Methyl (Z)-5,11,14,17-eicosatetraenoate	C_21_H_34_O_2_	318.5	Fatty acid methyl ester	1.31	Antibacterial [48]
21	32.06	Vitamin E	C_29_H_50_O_2_	430.7	Vitamins	0.65	Anti-cancer, hepatoprotective, and antispasmodic [36]
22	33.82	Campesterol	C_28_H_48_O	400.7	Phytosterol	0.83	Anti-inflammatory, antidiabetic, and anti-cancer [36]
23	34.56	Stigmasterol	C_29_H_48_O	412.7	Sterol	0.90	Anti-tumor, hypoglycemic, and anti-inflammatory [49,50,51]
24	34.65	Ursa-9(11),12-dien-3-ol	C_30_H_48_O	424.7	Triterpene	0.36	Anti-inflammatory, and antioxidant [52]
25	35.92	beta-Sitosterol	C_29_H_50_O	414.7	Phytosterol	1.54	Anti-cancer, and hypocholestremia [53]
26	36.09	Stigmasterol, 22,23-dihydro	C_29_H_50_O	414.7	Steroid	8.04	Anti-cancer, antioxidant, hypoglycemic, and anti-viral [54]
27	37.36	11-Oxours-12-en-3-yl acetate	C_32_H_50_O_3_	482.7	Ester of acetic acid	6.46	Antidiabetic [55]
28	36.73	beta-Amyrin	C_30_H_50_O	426.7	Triterpenoid	2.26	Antioxidant, antimalarial, and antiulcer [52]

RT: Retention time, M. formula: Molecular formula, % Area: % peak area, and Mol. wt: Molecular weight.

**Table 4 antibiotics-11-01155-t004:** Antibacterial activity of *F. vasta*.

	Zone of Inhibition (mm)				
Bacterial Strains	Conc. 25 mg/mL	Conc. 50 mg/mL	Conc. 75 mg/mL	Conc. 100 mg/mL	Standard Ceftriaxone 1 mg/mL
*Staphylococcus aureus*	18	20	21	22	26
*Staphylococcus epidermidis*	16	17	20	21	26
*Escherichia coli*	20	21	22	24	25
*Bordetella bronchiseptica*	12	13	15	18	20
*Bacillus subtilis*	12	16	17	19	22
*Bacillus pumilus*	13	14	16	17	21
*Micrococcus luteus*	15	17	18	20	21

**Table 5 antibiotics-11-01155-t005:** Antifungal activity of *F. vasta* by agar tube dilution method.

Sample	Fungal Strains	Linear Growth in Test Tubes (mm)	Linear Growth in Control (mm)	% Age Inhibition
*F. vasta*	*Aspergillus niger*	42	98	42%
	*Fusarium avenaceum*	50	91	54.94%
	*Fusarium brachygibbosum*	49	101	48.51%

**Table 6 antibiotics-11-01155-t006:** Anti-viral activity of *F. vasta*.

Strains	Titer Count in Control	Titer Count in Acyclovir	Titer Count in *F. vasta*
IBV	1024	00	00
NDV	2048	00	00
H9	2048	00	16

IBV; avian infectious bronchitis virus, H9; Influenza virus, and NDV; Newcastle disease virus, HA titer 0 to 8: highly strong, 16 to 32: strong, 64 to 128: moderate, and 256 to 2048: not active [56].

**Table 7 antibiotics-11-01155-t007:** Hemolytic activity of *F. vasta* and standard Triton X-100.

Extract/Standard	Hemolytic Activity %
*F. vasta*	6.39 ± 0.45
Triton X-100	92.27 ± 4.71

**Table 8 antibiotics-11-01155-t008:** Binding affinity and interactions of best-docked compounds.

Sr. No.	Ligand	Structures	α-Amylase			α-Glucosidase		
			Binding Energy	Amino Acids H-Bond Interactions	Amino Acids Hydrophobic Interactions	Binding Energy	Amino Acids H-Bond Interactions	Amino Acids Hydrophobic Interactions
1	Ursa- 9(11),12-dien-3-ol	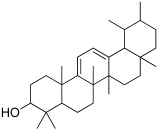	−10.5	ASP 197	TRP 59, HIS 101, LEU 162, TYR 62, ALA 198, TYR 62, THR 163, GLN 63, LEU 165	−8	-	PRO 223, PHE 225, LEU 219, GLU 141, GLN 392, TYR 388, LYS 290, ASP 289, TRP 288
2	Olean-12-en-3-ol, acetate, (3beta)-	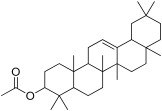	−10.4	-	TRP 59, GLU 233, ASP 356, HIS 305, TRP 58, ASP 300, THR 163, ASP 197, HIS 299, TYR 62, LEU 165, HIS 101	−8.1	LEU 287, ASN 258	PHE 225, GLU 226, VAL 222, PRO 214, PRO 223, GLY 286
3	beta-Amyrin	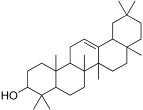	−10.1	GLU 233, ASP 197	TRP 59, THR 163, ASP 300, LEU 162, TYR 62, HIS 299, TRP 58	−8.4	GLU 113	TYR 122, PRO 109, TRP 110, GLU 183, GLU 180, TYR 68, LYS 69
4	11-Oxours-12-en-3-yl acetate	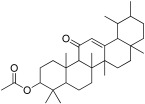	−9.6	-	TRP 58, GLU 233, ASP 197, TRP 59, LEU 162, TYR 62, THR 163	−8.1	-	LYS 395, TYR 221, LEU 219, GLU 141, GLN 392, PRO 223, PHE 225, ASP 289, TRP 288, LYS 290, ILE 391
5	Campesterol	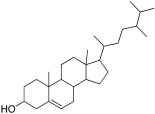	−9.3	GLU 240	TYR 151, HIS 299, ASP 197, LEU 162, ILE 235, ALA 198, HIS 201, TYR 62, GLU 233, TRP 59, TRP 58, LYS 200, ASP 300	−6.9	LYS 205	LYS 206, GLU 173, ASN 171, PHE 210, ILE 127, TRP 128, ASP 124, HIS 129, LYS 118, GLY 209, ALA 208
6	beta-Sitosterol	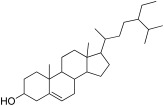	−9.3	GLU 240	ILE 235, HIS 201, TYR 62, TYR 151, HIS 299, TRP 58, ALA 198, ARG 195, ASP 197, LEU 162, TRP 59, LYS 200, GLU 233	−6.9	ASN 258	LYS 290, PRO 223, PHE 225, MET 229, LEU 287, ASP 289, TRP 288, SER 145, GLU 141, ILE 143
7	Stigmasterol	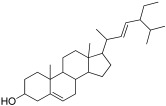	−9.1	GLU 233	ALA 198, TRP 59, LEU 162, ASP 197, LEU 165, GLN 63, GLY 104, THR 163	−7.5	GLN 392, LYS 395	MET 229, PHE 225, GLU 141, PRO 223, ILE 391, SER 145, TYR 388, LYS 290, TRP 288, LEU 287
8	Acarbose (standard)	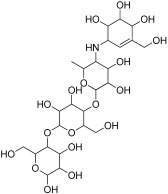	−7.7	ARG 252, GLY 403, ARG 421, THR 11, ARG 398, THR 6, GLN 404, GLY 334, SER 3	PRO 332, PRO 405, PHE 335, ASP 402, GLN 8	−6.7	ASN 58, ARG 17, ASP 59, ASN 61, ASP 381, ASP 379, LYS 436, PRO 433	VAL 383, TRP 434, MET 435, ASP 378,

**Table 9 antibiotics-11-01155-t009:** Lipinski rule of five and solubility of best-docked compounds.

Sr no.	Best-Docked Compounds	Lipinski’s Rule						Solubility		
		HBD	HBA	MWT	Lipophilicity	M.R	LR	ESOL Class	Ali Class	Silicos-IT Class
1	Ursa-9(11),12-dien-3-ol	1	1	424.7	4.81	134.67	1	PS	PS	PS
2	Olean-12-en-3-ol, acetate, (3beta)-	0	2	468.75	5.19	144.62	1	PS	IS	PS
3	beta-Amyrin	1	1	426.72	4.74	134.88	1	PS	PS	PS
4	11-Oxours-12-en-3-yl acetate	0	3	482.74	4.79	145.08	1	PS	PS	PS
5	Campesterol	1	1	400.68	4.92	131.23	1	PS	PS	MS
6	beta-Sitosterol	1	1	414.71	4.79	133.23	1	PS	PS	PS
7	Stigmasterol	1	1	412.69	5.01	132.75	1	PS	PS	MS

HBD; hydrogen bond doners, HBA; hydrogen bond acceptors, MWT; molecular weight, M.R; molar refractivity, LR; Lipinski rule, PS; poorly soluble, MS; moderate soluble, and IS; insoluble.

## Data Availability

Not applicable.

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
