# Peer review of "Phytochemical Profiling, In Vitro Biological Activities, and In-Silico Studies of Ficus vasta Forssk.: An Unexplored Plant"

_antibiotics, 2022, doi:10.3390/antibiotics11091155_

Round 1

Reviewer 1 Report

Research focused on determining the chemical composition and biological properties of poorly studied plants reveals potential therapeutic candidates, which drive drug discovery programs. This manuscript explores the phytochemistry and multifunctionality of the alcoholic extract of F. vasta. The results highlight the biomedical potential of specialized and inhibitory metabolites. However, the presentation of the work must be significantly improved for a high quality journal.

1. The title contradicts the introduction (lines 107-115).

2. Please review lines 21 and 22. This sentence should be rewritten considering the existence of previous studies.

3. The abstract is repetitive. The words biological and phytochemical appear several times in lines 22, 23, 24, 25 and 26. Review other sections as well. For example, in the introduction, the word medicinal (lines 43,44) and so on.

4.  Line 25. highest? What was the comparison made?

5. The authors showed a variety of promising biological activities, but did not evaluate the toxicity of this extract in normal or healthy cells in any assay. In this sense, it is not possible to discuss its selectivity and potential application.

6. Line 37. Good is a very generic term.

7. Line 47. Replace Due.. by...due

8. line 125. Rewrite is required.

9. The quality of figure 1 is very low and the identification of the axes should be clearer.

10. A brief description of tables and figures should be included with the titles to facilitate understanding.

11. The areas presented in table 3 were not described in the results, discussed and explored in the manuscript.

12. Retention time and molecular weight units should be included in table 3.

13. The authors express the same results in tables and graphs, without a clear meaning.

14. The order of figures should be revised. The authors presented two figures 4.

15. Figure 7 is of low quality and difficult to understand.

16. What are the standards, calibration curve and R2 for the quantitative determination of phenolic and flavonoid content?

17. What was the negative control used in the biological and enzymatic assays? Was the extract always dissolved in DMSO in the same manner as described for antibacterial assays?

18. Brand of reagents and model of equipment used must be clearly specified in the methodology section.

Author Response

Response to Respected Reviewer 1 comments

Response: Thank you for your kind observation. Also, thank you so much for your valuable insight on improving the manuscript. We greatly appreciate the reviewers’ comments and constructive suggestions. Those valuable comments are constructive to improve our manuscript and provide meaningful guidance for our future research. We hope this revised manuscript will meet the satisfaction of the reviewer. We again appreciate your helpful suggestions. If you have any further suggestions for changes, please let us know.

All the Manuscript including research design, methodology, references, results and conclusion has been improved according to reviewer suggestion.

Research focused on determining the chemical composition and biological properties of poorly studied plants reveals potential therapeutic candidates, which drive drug discovery programs. This manuscript explores the phytochemistry and multifunctionality of the alcoholic extract of F. vasta. The results highlight the biomedical potential of specialized and inhibitory metabolites. However, the presentation of the work must be significantly improved for a high quality journal.

As per your precious recommendation, we have made changes in our manuscript

  1. The title contradicts the introduction (lines 107-115).

Response: The authors acknowledge the expertise of reviewer and the title have been improved

  1. Please review lines 21 and 22. This sentence should be rewritten considering the existence of previous studies.

Response: These lines have been rewritten according to your suggestion.

  1. The abstract is repetitive. The words biological and phytochemical appear several times in lines 22, 23, 24, 25 and 26. Review other sections as well. For example, in the introduction, the word medicinal (lines 43,44) and so on.

Response: The abstract have been improved and repetitions have been removed in all the abstract

  1. Line 25. highest? What was the comparison made?

Response: This word have been removed and sentence is rephrased now.

  1. The authors showed a variety of promising biological activities, but did not evaluate the toxicity of this extract in normal or healthy cells in any assay. In this sense, it is not possible to discuss its selectivity and potential application.

Response: The authors agreed with your suggestion and now conducted the Hemolytic study for safety profile of the plant extract. The results and methodology is now mentioned in the manuscript.

  1. Line 37. Good is a very generic term.

Response: This word is corrected now

  1. Line 47. Replace Due.. by...due

Response: The authors have now corrected according to the suggestion.

  1. line 125. Rewrite is required.

Response: The specified line is rephrased

  1. The quality of figure 1 is very low and the identification of the axes should be clearer.

Response: The quality of figure is improved now

  1. A brief description of tables and figures should be included with the titles to facilitate understanding.

Response: Some description of tables and figures have been added. Now the figures and tables are self-explanatory.

  1. The areas presented in table 3 were not described in the results, discussed and explored in the manuscript.

Response: Now the areas are mentioned and discussed in the text of manuscript

  1. Retention time and molecular weight units should be included in table 3.

Response: These units are mentioned in the table

  1. The authors express the same results in tables and graphs, without a clear meaning.

Response: Now extra tables have been removed

  1. The order of figures should be revised. The authors presented two figures 4.

Response: The whole manuscript is read carefully and nearly all the flaws have been rectified.

  1. Figure 7 is of low quality and difficult to understand.

Response: The quality of Figure 7 is tried to improve.

  1. What are the standards, calibration curve and R2 for the quantitative determination of phenolic and flavonoid content?

Response: The TPC and TFC of F. vasta extract was carried out with the help of regression equation that was obtained from gallic acid and quercetin standard curve.

  1. What was the negative control used in the biological and enzymatic assays? Was the extract always dissolved in DMSO in the same manner as described for antibacterial assays?

Response: The negative control and positive control used in all biological activities are now mentioned in the manuscript. DMSO was used as solvent only in antimicrobial studies.

  1. Brand of reagents and model of equipment used must be clearly specified in the methodology section.

Response: Now all the brands of reagent and equipment is mentioned in the manuscript and highlighted.

Reviewer 2 Report

General comment: Reject

In my opinion, this manuscript should be rejected. There are too many odd or incorrect terms in the paper. English, throughout the article, needs to be corrected by a native speaker or an excellent linguist who knows the specificities of scientific English. There is a mess in vocabulary (e.g. words “anti-oxidant” and “antioxidant”, “anti-inflammatory” and “anti-inflammatory” are in the same table 3). In my opinion, a single style should apply throughout the text. In addition, references have not been prepared according to the guidelines required by the publisher (e.g. 20, 21, 22, 60, 72, 78, etc.). I also have concerns about the legibility of the tables; all should be improved. In that class of journals as Antibiotics, all captions to tables and figures should be self-explanatory. I suggest new submission after amendments.

Detailed comments:

Abstract

Line 28                 instead of “related with” should be “related to”

Line 32                 should be “the extract”

Line 37                 add “ethanol extract of F. vasta”

Introduction:

Line 43                 “novel lead compounds” check english

Line 45                 “ailments” are the wrong word in this context

Line 48                 “innovative lead compounds” as above check english

Line 47                 change for Recently due to the

Line 51                 should be “is a metabolic disease”

Line 62                 it seems that hyphens are missing in words: “α amylase” and “α glucosidase”

Line 64               rephrases this sentence

Line 67, 95          citations are needed

Line 72                 change “scavengers for free radical” for “scavengers of free radical” or “free radical scavengers”

Line 73                 “external influences” check english

Results:

Line 125              “Table 1. Showed…” change to lower case letter

Line 131              “table 2” change to upper case letter

Line 143              this sentence appears to be part of the description of Table 2

Line 158              a space is missing between Table 3 and Phytocompounds

Line 176, 178     lower case letter in acarbose

Line 185              change the first letter in the word “percentage” to capital (start of description in Fig. 3)

Line 200        in my opinion, this sentence: “The results are shown in Table 6, and they showed that F.vasta has antifungal activity against these three strains” is misleading and should be rephrased.

Line 204        spaces are missing between: ,i.e.,

Line 214        “Table 8” with a capital letter

Line 236, 239     hyphens are missing in word beta amyrin

Line 238              space is missing before “The docking results”

Discussion:

Line 291              turn to bold

Line 293              a space is missing before the word “flavonoids”

Line 307              different instead of dffrent

Line 315, 317     hyphens are missing in the words “α glucosidase” and “α amylase” (single style throughout the whole text)

Line 319              a space is missing in “in vitro”

Line 320              Ficus vasta turn to italic

Line 326              hyphens are missing in the words “gram positive” and “gram negative”

Line 337        rephrases the sentence as follows: "Tentative identification of the ethanolic extract by GCMS revealed many compounds with antibacterial, antifungal, and antiviral activities, namely"

Line 341, 342     all names of microorganisms should be italicised

Line 351              a space is missing after the word “cancer”

Materials and Methods:

Line 395              add a space before “saponins”

Line 396              add a space between “was determined”

Line 400              upper case in the name of “Folin-Ciocalteu method”

Line 412              it seems to be a double space before the word "solution"

Line 419              vasta should be written in lower case

Line 457              add a space before In

Line 520         change the word “Haemagglution” into “Haemagglutionation”, “the R.B.Cs” abbreviation should be explained

Line 550          add “with” before added 10% fetal bovine serum

Line 551           turn 2 to lower index in carbon dioxide

Line 554           add some space between activity and as described

Line 555            should be “modifications”

Line 577            remove spaces in brackets with the link

Line 580            a space is missing between to and 10

Line 585             add spaces between mean ± SD

Conclusions:

Line 592              something is missing in the word “antidiabetic”

Table 1 Sr. No. 4     Flavonoids seem to be the more popular name

Table 3                    Mess in vocabulary and style (some words with the upper case, some with the lower case). Some standardisation is needed.

Figure 3                  change the alpha symbols on the graph (the current ones are blurred)

Figure 4                  names of microorganisms are hard to read; the figure must be improved

Figure 5                  Improve axis descriptions

Figure 6, 7              to maintain one style change A-amylase into α-amylase, and A-glucosidase into α-glucosidase

Figure 7                  poor quality, blurred, must be improved

Figure 8                  there are two figures 7 in the manuscript; the following figures should be renumbered

Author Response

Response to Respected Reviewer 2 Comments

In my opinion, this manuscript should be rejected. There are too many odd or incorrect terms in the paper. English, throughout the article, needs to be corrected by a native speaker or an excellent linguist who knows the specificities of scientific English. There is a mess in vocabulary (e.g. words “anti-oxidant” and “antioxidant”, “anti-inflammatory” and “anti-inflammatory” are in the same table 3). In my opinion, a single style should apply throughout the text. In addition, references have not been prepared according to the guidelines required by the publisher (e.g. 20, 21, 22, 60, 72, 78, etc.). I also have concerns about the legibility of the tables; all should be improved. In that class of journals as Antibiotics, all captions to tables and figures should be self-explanatory. I suggest new submission after amendments.

Response: Thank you for your kind observation. Also, thank you so much for your valuable insight on improving the manuscript. Now all the manuscript have been read again carefully and English have been improved. Also uniformity in the text have been brought. The tables and figures along with their titles have been improved. We hope that this revised version of manuscript will be according to your expectation. Those valuable comments are constructive to improve our manuscript and provide meaningful guidance for our future research.

Kindly reconsider our manuscript for revision.

Detailed comments:

Abstract

Line 28                 instead of “related with” should be “related to”

Line 32                 should be “the extract”

Line 37                 add “ethanol extract of F. vasta”

Response: The authors acknowledge your expertise and all the required changes in the abstract have been performed.

Introduction:

Line 43                 “novel lead compounds” check english

Line 45                 “ailments” are the wrong word in this context

Line 48                 “innovative lead compounds” as above check english

Line 47                 change for Recently due to the

Line 51                 should be “is a metabolic disease”

Line 62                 it seems that hyphens are missing in words: “α amylase” and “α glucosidase”

Line 64               rephrases this sentence

Line 67, 95          citations are needed

Line 72                 change “scavengers for free radical” for “scavengers of free radical” or “free radical scavengers”

Line 73                 “external influences” check English

Response: The authors are very thankful for guiding in such a way for the improvement of manuscript. All required changes were made according to your suggestions in the introduction and now it looked very improved.

Results:

Line 125              “Table 1. Showed…” change to lower case letter

Line 131              “table 2” change to upper case letter

Response: This is changed accordingly

Line 143              this sentence appears to be part of the description of Table 2

Response: Yes this is foot note of Table 2 and explained further

Line 158              a space is missing between Table 3 and Phytocompounds

Response: Space is now added. This error occurred in most places due to old version of micro soft office. That have deleted many spaces in many places and caused a lot of trouble for reviewer. No we have rectified the problem.

Line 176, 178     lower case letter in acarbose

Line 185              change the first letter in the word “percentage” to capital (start of description in Fig. 3)

Response: The change is done according to your suggestions

Line 200        in my opinion, this sentence: “The results are shown in Table 6, and they showed that F.vasta has antifungal activity against these three strains” is misleading and should be rephrased.

Response: This line is rephrased and now looked improved.

Line 204        spaces are missing between: ,i.e.,

Response: A space is added in required place.

Line 214        “Table 8” with a capital letter

Response: The change is performed accordingly.

Line 236, 239     hyphens are missing in word beta amyrin

Response: Now hyphens are added in the advised places

Line 238              space is missing before “The docking results”

Discussion:

Line 291              turn to bold

Line 293              a space is missing before the word “flavonoids”

Response: Instructions are followed accordingly

Line 307              different instead of dffrent

Response: This is now corrected

Line 315, 317     hyphens are missing in the words “α glucosidase” and “α amylase” (single style throughout the whole text)

Response: Now hyphens are added

Line 319              a space is missing in “in vitro”

Line 320              Ficus vasta turn to italic

Response: Space is added in the mentioned place and plant name is made italic

Line 326              hyphens are missing in the words “gram positive” and “gram negative”

Response: Hyphens are added

Line 337        rephrases the sentence as follows: "Tentative identification of the ethanolic extract by GCMS revealed many compounds with antibacterial, antifungal, and antiviral activities, namely"

Response: This sentence was rephrased according to instructions given.

Line 341, 342     all names of microorganisms should be italicized

Response: All name of microorganism are made italic

Line 351              a space is missing after the word “cancer”

Response: Space is added now

Materials and Methods:

Line 395              add a space before “saponins”

Line 396              add a space between “was determined”

Response: Spaces were added in both these places in the text

Line 400              upper case in the name of “Folin-Ciocalteu method”

Response: Change was performed according to the suggestion

Line 412              it seems to be a double space before the word "solution"

Response: Change was done and double space was removed

Line 419              vasta should be written in lower case

Response: This was changed in lower case

Line 457              add a space before In

Response: Space was added

Line 520         change the word “Haemagglution” into “Haemagglutionation”, “the R.B.Cs” abbreviation should be explained

Response: The change in the text is made accordingly and the R.B.Cs are explained.

Line 550          add “with” before added 10% fetal bovine serum

Response: This was added in the specified place.

Line 551           turn 2 to lower index in carbon dioxide

Response: The change was performed accordingly

Line 554           add some space between activity and as described

Line 555            should be “modifications”

Line 577            remove spaces in brackets with the link

Line 580            a space is missing between to and 10

Line 585             add spaces between mean ± SD

Response: All the advised changes have been performed accordingly.

Conclusions:

Line 592              something is missing in the word “antidiabetic”

Response: Now this word is corrected.

Table 1 Sr. No. 4     Flavonoids seem to be the more popular name

Response: The change is performed accordingly

Table 3                    Mess in vocabulary and style (some words with the upper case, some with the lower case). Some standardisation is needed.

Response: The authors acknowledge the observations of reviewer and now all the table has been modified and it looks improved and uniform.

Figure 3                  change the alpha symbols on the graph (the current ones are blurred)

Response: These symbols are revised

Figure 4                  names of microorganisms are hard to read; the figure must be improved

Response: The picture quality was tried to improve

Figure 5                  Improve axis descriptions

Response: The axis description of the figure have been improved (Figure 6 now)

Figure 6, 7              to maintain one style change A-amylase into α-amylase, and A-glucosidase into α-glucosidase

Response: This have been changed (Figure 7 and 8 now in revised MS)

Figure 7                  poor quality, blurred, must be improved

Response: This have been tried to improve the resolution of that specified picture

Figure 8                  there are two figures 7 in the manuscript; the following figures should be renumbered

Response: The numbering for all the figures have been checked and corrected.

Reviewer 3 Report

Dears authors

1 - Introduction: the contents and the drafting of the general part must be reformed to review the syntax of the topic

3- Discussion: to deepen in view of the problem of antibiotic resistance through the use of natural product against multidrug-resistant strains of clinical relevance. Learn more about this by using and citing the following references (optional): PMID: 35456809 ; PMID: 35684205 ; PMID: 35736695

3 - Check the bibliographic entries throughout the text, some of which are non-compliant.

4 - Revision of English grammar and in particular of applied scientific English: in particular verb tenses and syntax in the discussion

Author Response

Response to Respected Reviewer 3 comments

Thank you for your kind observation. Also, thank you so much for your valuable insight on improving the manuscript. We greatly appreciate the reviewers’ comments and constructive suggestions. Those valuable comments are constructive to improve our manuscript and provide meaningful guidance for our future research. We hope this revised manuscript will meet the satisfaction of the reviewer. We again appreciate your helpful suggestions. If you have any further suggestions for changes, please let us know.

Introduction, research design, methodology, references, results and conclusion of the manuscript has been improved according to reviewer suggestion.

Dears authors

1 - Introduction: the contents and the drafting of the general part must be reformed to review the syntax of the topic

Response: The authors acknowledge the concerns of reviewer and now the introduction part is revised and improved.

3- Discussion: to deepen in view of the problem of antibiotic resistance through the use of natural product against multidrug-resistant strains of clinical relevance. Learn more about this by using and citing the following references (optional): PMID: 35456809 ; PMID: 35684205 ; PMID: 35736695

Response: The authors have improved the discussion and the reference suggested has been incorporated in the revised manuscript

3 - Check the bibliographic entries throughout the text, some of which are non-compliant.

Response: The authors have checked bibliography and modified where required.

4 - Revision of English grammar and in particular of applied scientific English: in particular verb tenses and syntax in the discussion

Response: The authors have read the Manuscript very carefully and now removed all the mistakes

Reviewer 4 Report

The full name of the species (Ficus vasta Forssk.) should be used in the title and with the first use in the abstract and full-text.

 Line 45: „to cure“ means „to make someone with an illness healthy again“. Unfortunately, currently there is no medicine or medicinal plant able to cure asthma, diabetes or other of the diseases mentioned there. The verb used should be “to treat”, which does not necessarily imply a cure. The same holds true for line 63: diabetes is not a curable disease.  

Lines 59-61: the claim that “α-amylase and α-glucosidase inhibitors are the most beneficial in diabetes” is simply not evidence-based. They are rather among the least important, and clinical guidelines from different geopolitical regions do not recommend them as first line options or as the most important (of contrary, their place in the treatment of diabetes tends to be controversial or difficult to find).

Lines 61-63: “antioxidants” is a very wide group, not all antioxidants inhibit alpha-amylase or alpha-glucosidase (for instance ascorbic acid or tocopherol are antioxidants, but as far as we know, they are not such inhibitors).

Lines 69-70: that statement is also simply not true. It is valid that many plants/phytochemicals have been used as starting points for developing antitumour medicines (e.g. paclitaxel, docetaxel, vinca alkaloids), but they are far from being free from adverse effects, or even having superior safety.

Line 391: please state the name of the rotary evaporator and the degree of concentration (was the final extract a dried one? When was the concentration stopped?)

Lines 400-417, 432-454: please state the name of the spectrophotometer used and for all experiments the source of reagents.

Lines 483-491: please state the final concentration of DMSO in the samples. The same for line 503. The authors should also clarify the method and software used to estimate IC50 for the different antioxidant tests.

Lines 501-502: “A fungus piece of diameter  4  mm  was  used  to  inoculate  each  tube  removed  from  the  fungus  culture”. This sentence is unclear and needs to be rephrased for clarity.

Lines 504-506: were are not convinced that the authors estimated “age inhibition”. Why age? (if percentage was intended, it should be written as such; or only %). The same for Table 4.

Lines 508-518: please clarify the viral species used and their sources.

Lines 553-559: it would have been highly preferrable to also use a normal cell line for the assessment of cell viability.

Line 573: “Furthermore,  the  binding  affinity  values  yielded  the  same  results.” Considering that docking is a highly stochastic process, it is difficult to understand how it yielded “the same results”. Please rephrase this sentence to allow a better understanding of its meaning.

Lines 586-587: if one-way ANOVA was used, then it is highly likely that a post-hoc test was also used and the authors should clarify what such test was applied.  

Lines 140-142: we tend to be highly skeptical of the flavonoid contents measured, which are equivalent to 12.92% (very atypical, extremely large). Moreover, the flavonoid concentration is higher than the total polyphenolic concentration, which is a little strange, because usually flavonoids (only a subgroup of all polyphenols) tend to be in inferior amounts to those of polyphenols (the concentration of total polyphenols seems credible).

Table 3: We difficulties in understanding “Biomass burning proxies” as a biological activity.

Figure 3: the alpha-letter in the title needs to be corrected.

Lines 486, 499, and Tables 5 and 6: did the authors use concentrations as high as 100 mg/ml? Or were they ug/ml? Because 100 mg/ml are extremely high and MICs typically are expressed in ug/ml.

 Line 215: “47.78±4.21 was comparable to streptokinase 81.86±6.37”. The first value is actually about the half of the second.

Section 2.2.7 – an IC50 value should be estimated through non-linear regression.

Section 2.3.1 – the authors should provide proof that the docking process was valid (e.g. statistical measures for the accuracy of docking of known compounds, images of one or several docked reference substances against crystalopgraphic data etc).

The Discussion section is a single very long paragraph. It should be split in several paragraphs.  

Lines 324-330 – this discussions are only valid provided the concentrations used are relevant. “Although values lower than 16 mg/mL have been considered sometimes as showing a strong antibacterial effect [135], other authors have used more stringent criteria: MIC values < 100 μg/mL have been proposed to be highly active, those between 100 and 500 μg/mL active, those between 500 and 1000 μg/mL moderately active, those between 1000 and 200 μg/mL of low activity, and those with MIC > 2000 μg/mL inactive [136,137].” (https://www.ncbi.nlm.nih.gov/pmc/articles/PMC7761148/) Therefore this discussion should first consider the concentration levels used in the assessment of the antibacterial and antifungal effects.

Lines 331-333: because the antiviral effects did not measure directly this purported effect, but only indirectly, the limitations of the methods used should be discussed here.

Lines 358-382: this fragment to a good extent repeats information from the in silico-results. It would be preferrable here to discuss the limitations of the methods used, as well as potential wet-lab experiment confirmation for some of the compounds known to be present in the extract, if available. Otherwise, a short presentation of conclusion and a discussion of the limitations would be much preferrable in our view.

Author Response

Response to Respected Reviewer 4 comments

Response: Thank you for your kind observation. Also, thank you so much for your valuable insight on improving the manuscript. We greatly appreciate the reviewers’ comments and constructive suggestions. Those valuable comments are constructive to improve our manuscript and provide meaningful guidance for our future research. We hope this revised manuscript will meet the satisfaction of the reviewer.

Research design, methodology, references, results and conclusion of the manuscript has been improved according to reviewer suggestion.

The full name of the species (Ficus vasta Forssk.) should be used in the title and with the first use in the abstract and full-text.

Response: The full name of the plant has been added in the title, abstract and introduction.

 Line 45: „to cure“ means „to make someone with an illness healthy again“. Unfortunately, currently there is no medicine or medicinal plant able to cure asthma, diabetes or other of the diseases mentioned there. The verb used should be “to treat”, which does not necessarily imply a cure. The same holds true for line 63: diabetes is not a curable disease.

Response: The authors are highly thankful for very valuable comments for the improvement of manuscript and now all the required changes have been done in the manuscript

Lines 59-61: the claim that “α-amylase and α-glucosidase inhibitors are the most beneficial in diabetes” is simply not evidence-based. They are rather among the least important, and clinical guidelines from different geopolitical regions do not recommend them as first line options or as the most important (of contrary, their place in the treatment of diabetes tends to be controversial or difficult to find).

Response: The authors have now modified the sentence by removing the most beneficial and now the sentence looked improved.

Lines 61-63: “antioxidants” is a very wide group, not all antioxidants inhibit alpha-amylase or alpha-glucosidase (for instance ascorbic acid or tocopherol are antioxidants, but as far as we know, they are not such inhibitors).

Response: The authors have revised according to your prestigious suggestion.

Lines 69-70: that statement is also simply not true. It is valid that many plants/phytochemicals have been used as starting points for developing antitumour medicines (e.g. paclitaxel, docetaxel, vinca alkaloids), but they are far from being free from adverse effects, or even having superior safety.

Response: The authors acknowledge your suggestion and now amendment have been performed.

Line 391: please state the name of the rotary evaporator and the degree of concentration (was the final extract a dried one? When was the concentration stopped?)

Response: The rotary model is mentioned in the manuscript and degree of concentration is also written and highlighted.

Lines 400-417, 432-454: please state the name of the spectrophotometer used and for all experiments the source of reagents.

Response: The names of spectrophotometer has been added in all experiments

Lines 483-491: please state the final concentration of DMSO in the samples. The same for line 503. The authors should also clarify the method and software used to estimate IC50 for the different antioxidant tests.

Response: The authors have mentioned the concentration of DMSO in the manuscript. We calculated the IC50 values by first plotting the standard curve and by using the parameters of regression equation, this value was calculated.

Lines 501-502: “A fungus piece of diameter  4  mm  was  used  to  inoculate  each  tube  removed  from  the  fungus  culture”. This sentence is unclear and needs to be rephrased for clarity.

Response: This sentence was modified now.

Lines 504-506: were are not convinced that the authors estimated “age inhibition”. Why age? (if percentage was intended, it should be written as such; or only %). The same for Table 4.

Response: The change has been performed accordingly.

Lines 508-518: please clarify the viral species used and their sources.

Response: These were clarified and the source of strains are mentioned in manuscript.

Lines 553-559: it would have been highly preferrable to also use a normal cell line for the assessment of cell viability.

Response: The authors highly acknowledge your suggestion. But our aim was to investigate the anti-cancer activity of our plant extract.

Line 573: “Furthermore, the binding affinity values yielded the same results.” Considering that docking is a highly stochastic process, it is difficult to understand how it yielded “the same results”. Please rephrase this sentence to allow a better understanding of its meaning.

Response: The authors are thankful for such a deep revision of each and every aspect of manuscript. You are absolutely right that binding affinity obtained from these different software are not same. But we used AutoDock and AutoDock Vina embedded in PyRx by using same dimensions of grid box. That is why nearly similar results were obtained in terms of binding affinity. There was no significant difference among these results.

Lines 586-587: if one-way ANOVA was used, then it is highly likely that a post-hoc test was also used and the authors should clarify what such test was applied.

Response: The one-way ANOVA was calculated by post-hoc test and mentioned in the manuscript.

Lines 140-142: we tend to be highly skeptical of the flavonoid contents measured, which are equivalent to 12.92% (very atypical, extremely large). Moreover, the flavonoid concentration is higher than the total polyphenolic concentration, which is a little strange, because usually flavonoids (only a subgroup of all polyphenols) tend to be in inferior amounts to those of polyphenols (the concentration of total polyphenols seems credible).

Response: The reviewer observation is acknowledged. But we presented the results which we obtained during experimentation. Also there are evidences in the literature that flavonoids may be higher than phenolics (doi:10.3390/ijms13032707 and doi.org/10.1016/j.sajb.2022.04.038)

Table 3: We difficulties in understanding “Biomass burning proxies” as a biological activity.

Response: Biomass burning proxies is actually general use of this compound, and now it has been replaced with antibacterial activity according to literature.

Figure 3: the alpha-letter in the title needs to be corrected.

Response: The alpha-letter was modified now.

Lines 486, 499, and Tables 5 and 6: did the authors use concentrations as high as 100 mg/ml? Or were they ug/ml? Because 100 mg/ml are extremely high and MICs typically are expressed in ug/ml.

Response: The authors really acknowledge the expertise of reviewer. But our intent was to calculate the zone of inhibition only. We want to check the antibacterial potential of plant either it have or not. The plant extracts are mixture of compounds, among them some may have antibacterial potential that is why for the estimation of plant extract we have used concentrated solution of extract.

 Line 215: “47.78±4.21 was comparable to streptokinase 81.86±6.37”. The first value is actually about the half of the second.

 Response: This sentence has been rephrased in the text.

Section 2.2.7 – an IC50 value should be estimated through non-linear regression.

Response: The IC50­ value is calculated and now mentioned in manuscript.

Section 2.3.1 – the authors should provide proof that the docking process was valid (e.g. statistical measures for the accuracy of docking of known compounds, images of one or several docked reference substances against crystalopgraphic data etc).

Response: The authors have mentioned the validation of docking studies. The pictures of reference compounds (acarbose) have been provided in the manuscript.

The Discussion section is a single very long paragraph. It should be split in several paragraphs. 

Response: The change has been performed accordingly.

Lines 324-330 – this discussions are only valid provided the concentrations used are relevant. “Although values lower than 16 mg/mL have been considered sometimes as showing a strong antibacterial effect [135], other authors have used more stringent criteria: MIC values < 100 μg/mL have been proposed to be highly active, those between 100 and 500 μg/mL active, those between 500 and 1000 μg/mL moderately active, those between 1000 and 200 μg/mL of low activity, and those with MIC > 2000 μg/mL inactive [136,137].” (https://www.ncbi.nlm.nih.gov/pmc/articles/PMC7761148/) Therefore this discussion should first consider the concentration levels used in the assessment of the antibacterial and antifungal effects.

Response: This very skilled comment represents the very deep observation and knowledge of reviewer. The plant extracts are mixture of a lot of compounds. Some compounds may have antibacterial activity and some may have no antibacterial activity. Therefore the extract concentration used in the assay was relatively high as compared to pure antibacterial agents.  

Lines 331-333: because the antiviral effects did not measure directly this purported effect, but only indirectly, the limitations of the methods used should be discussed here.

Response: The antiviral activity was performed by Heamagglutination (HA) test. This test is not used for the assay against Infectious Bursal Disease Virus. This method has also least sensitivity.

Lines 358-382: this fragment to a good extent repeats information from the in silico-results. It would be preferrable here to discuss the limitations of the methods used, as well as potential wet-lab experiment confirmation for some of the compounds known to be present in the extract, if available. Otherwise, a short presentation of conclusion and a discussion of the limitations would be much preferrable in our view.

Response: The authors acknowledge the reviewer suggestion. The discussion part of in-silico studies is improved now.

Round 2

Reviewer 1 Report

Authors have addressed most of my previous comments. However, despite the efforts made, some important points still need to be reviewed.

1. Line 44. diseases

2. Quality of figure 1 is still not appropriate.What are the units of the axes in the figure?

3. Figure 4 represents the same information as in Table 4 and may be included in supplementary material.

4. Authors should standardize the colors of the figures in order to facilitate understanding. In the present version, the colors do not bring significant information and confuse the reader.

5. The authors added hemolytic assays as a reference for the study of toxicity. Although widely used in the early stages of drug development, erythrocytes are a cellular model with limitations for the comparisons performed. Authors should discuss this or add assays with another cell type. Why did the authors express hemolysis as a percentage and not calculate the HC50 as they did for most other in vitro biological assays?

6. The hemolysis results presented contradict the presented methodology. How was the percentage of hemolysis calculated? Wouldn't the absorbance of the positive control be considered as 100% hemolytic activity?

7. Line 235. IC50 should be expressed in the same units as in the figure.

8. Figure 9 remains low quality.

9. Authors should reduce the number of figures and tables focusing on the main results of the investigation. Some results can be presented in the same figure and section, and others can be added to supplementary material.

10. Provide details about concentrations of calibrations curves, R2 and regression factors for quantitative analysis of secondary metabolites.

Author Response

Authors have addressed most of my previous comments. However, despite the efforts made, some important points still need to be reviewed.

Response: Thank you for your kind observation. Also, thank you so much for your valuable insight on improving the manuscript. We greatly appreciate the reviewers’ comments and constructive suggestions. Those valuable comments are constructive to improve our manuscript and provide meaningful guidance for our future research. We hope this revised manuscript will meet the satisfaction of the reviewer. We again appreciate your helpful suggestions. If you have any further suggestions for changes, please let us know.

  1. Line 44. diseases

Response: The authors are thankful for highlighting the mistakes to improve the manuscript. Now this word has been corrected.

  1. Quality of figure 1 is still not appropriate. What are the units of the axes in the figure?

Response: The authors tried their best to improve the quality of specified figure. But this is software generated image. The resolution of this figure cannot be improved further. The units of axes are described in manuscript

  1. Figure 4 represents the same information as in Table 4 and may be included in supplementary material.

Response: This figure is now moved to supplementary file

  1. Authors should standardize the colors of the figures in order to facilitate understanding.In the present version, the colors do not bring significant information and confuse the reader.

Response: The observation of reviewer is very right. Now the color of figures has been improved and brought uniformity in the color.

  1. The authors added hemolytic assays as a reference for the study of toxicity.Although widely used in the early stages of drug development, erythrocytes are a cellular model with limitations for the comparisons performed. Authors should discuss this or add assays with another cell type. Why did the authors express hemolysis as a percentage and not calculate the HC50 as they did for most other in vitro biological assays?

Response: The authors have performed only in vitro biological activities and only hemolytic assay for safety profiling of the plant. The authors will perform other toxicological activities before going to in vivo study. The limitations of hemolytic activity is now mentioned in the manuscript. The observation of respected reviewer is right that in most studies reported, we used IC50 methodology. But in some cases we used % inhibition methods; likely in thrombolytic and in antifungal activity. And for performing this activity, we followed the paper which also describes the results in % hemolytic activity.

  1. The hemolysis results presented contradict the presented methodology. How was the percentage of hemolysis calculated?Wouldn't the absorbance of the positive control be considered as 100% hemolytic activity?

Response: The hemolytic activity was determined according to the literature cited in the text. The authors acknowledge the reviewer comment. There are other formulas also available for calculation of hemolytic activity. We used the formula which is described in the literature cited in the text. The standard agent used in the assay revealed maximum hemolytic activity. Therefore, mostly in the literature, the positive control is written as 100 % hemolytic active.

  1. Line 235. IC50 should be expressed in the same units as in the figure.

Response: Thanks for highlighting the mistake and now this is corrected.

  1. Figure 9 remains low quality.

Response: The quality of this figure is improved.

  1. Authors should reduce the number of figures and tables focusing on the main results of the investigation.Some results can be presented in the same figure and section, and others can be added to supplementary material.

Response: The authors have now shifted some figures and tables to the supplementary file, following your suggestion.

  1. Provide details about concentrations of calibrations curves, R2 and regression factors for quantitative analysis of secondary metabolites.

Response: The detail about calibration curve have been added.

Reviewer 2 Report

Line 35                 add the” before “ethanol extract of F. vasta”

Line 44                 replace “diseses” with “diseases”          

Line 47                 replace “to produce” with “to obtain”

Line 69                 “Antioxidants are compounds that act as scavengers of free radicals created by the body's cells due to various metabolic reactions and external factors because these free radicals are unstable, they cause serious damage to the proteins, lipids, DNA, and RNA in the human body. These free radicals are one of the most common causes of cancer in the human body. These antioxidants are useful to cells because they help to prevent the growth of cancer.”

                               Numerous repetitions of words nearby - improve the style of this part of the text.

Line 137, 158      Different font sizes- should be corrected

Line 155               “compounds”

Line 189-196       some additional information seems to be missing. What type of antibacterial effect is described here? Growth inhibition? Correct this part of the text.

Line 251                should be “exhibit”

Line 270                 In paragraph 2.3.2 (ADMET analysis), the word “compound” was used excessively. This part of the text should be reworded.

Line 272                 should be “physicochemical properties”

Line 314, 331        should be “flavonoid compounds”

Line 363                  remove the percentage symbol

Line 396                hyphen is missing in “pi alkyl”, replace “protein-ligan” with “protein-ligand”

Line 410               should be the “blood-brain barrier”

Line 417                correct the TPSA unit (it should be A2)

Line 593                remove “test procedure”

Line 649               should be “Open Babel”

References          The references still have some errors. It must be improved.

Table 3                 Sr No. 17 align the spelling of antimicrobial

Figure 3               correct the alpha symbols on the graph (the current ones are still blurred and cannot be approved)

Figure 9               low quality, blurred, and currently cannot be approved

Author Response

Response to Respected Reviewer 2 comments

Thank you for your kind observation. Also, thank you so much for your valuable insight on improving the manuscript. We greatly appreciate the reviewers’ comments and constructive suggestions. Those valuable comments are constructive to improve our manuscript and provide meaningful guidance for our future research. We hope this revised manuscript will meet the satisfaction of the reviewer. We again appreciate your helpful suggestions. The changes performed in this round of revision were highlighted with green color.

Line 35                 add “the” before “ethanol extract of F. vasta”

Response: This added in the advised place

Line 44                 replace “diseses” with “diseases”          

Response: The authors are thankful for highlighting the mistakes to improve the quality of manuscript. Now it has been corrected.

Line 47                 replace “to produce” with “to obtain”

Response: The change is done according to the precious suggestion of the reviewer

Line 69                 “Antioxidants are compounds that act as scavengers of free radicals created by the body's cells due to various metabolic reactions and external factors because these free radicals are unstable, they cause serious damage to the proteins, lipids, DNA, and RNA in the human body. These free radicals are one of the most common causes of cancer in the human body. These antioxidants are useful to cells because they help to prevent the growth of cancer.”

                               Numerous repetitions of words nearby - improve the style of this part of the text.

Response: This paragraph is rephrased for improvement.

Line 137, 158      Different font sizes- should be corrected

Response: This is corrected in size.

Line 155               “compounds”

Response: The required change is brought in the manuscript

Line 189-196       some additional information seems to be missing. What type of antibacterial effect is described here? Growth inhibition? Correct this part of the text.

Response: Some detail is added and highlighted

Line 251                should be “exhibit”

Response: This is changed according to the suggestion of the reviewer.

Line 270                 In paragraph 2.3.2 (ADMET analysis), the word “compound” was used excessively. This part of the text should be reworded.

Response: This paragraph was rephrased.

Line 272                 should be “physicochemical properties”

Line 314, 331        should be “flavonoid compounds”

Line 363                  remove the percentage symbol

Line 396                hyphen is missing in “pi alkyl”, replace “protein-ligan” with “protein-ligand”

Line 410               should be the “blood-brain barrier”

Line 417                correct the TPSA unit (it should be A2)

Line 593                remove “test procedure”

Line 649               should be “Open Babel”

Response: The authors are extremely thankful for such a professional review of their manuscript. All the changes advised are performed in the manuscript and are highlighted

References          The references still have some errors. It must be improved.

Response: This portion have been improved.

Table 3                 Sr No. 17 align the spelling of antimicrobial

Response: This have been corrected.

Figure 3               correct the alpha symbols on the graph (the current ones are still blurred and cannot be approved)

Response: The authors are thankful and have improved the graph according the suggestion of reviewer

Figure 9               low quality, blurred, and currently cannot be approved

Response: The quality of this figure is improved and now it looks clear.

Reviewer 3 Report

Been made the corrections

Author Response

The authors are highly thankful for appreciating the revised manuscript. Also, thank you so much for your valuable insight for the improvement of the manuscript. The English language of manuscript and the result section has been improved.

Reviewer 4 Report

A number of improvements have been made. However, with respect to docking, the validation evidence is very scarce if at all. What the authors should show is a comparison of the crystallographic pose of acarbose supperposed over the docked pose (see e.g. https://www.researchgate.net/profile/Douglas-Houston-2/publication/235373295/figure/fig2/AS:299705239130115@1448466705480/Comparison-of-docking-and-crystallographic-binding-poses-in-the-case-of-a-false-positive_W640.jpg or https://www.researchgate.net/profile/Graziela-Heberle-2/publication/50265549/figure/fig7/AS:394356612714502@1471033351621/Fig-10-Superposition-of-the-best-docked-structure-and-crystallographic-structure-for_W640.jpg) . I am not convinced that with the current way of reporting the docking proves anything.

Author Response

A number of improvements have been made. However, with respect to docking, the validation evidence is very scarce if at all. What the authors should show is a comparison of the crystallographic pose of acarbose supperposed over the docked pose (see e.g. https://www.researchgate.net/profile/Douglas-Houston-2/publication/235373295/figure/fig2/AS:299705239130115@1448466705480/Comparison-of-docking-and-crystallographic-binding-poses-in-the-case-of-a-false-positive_W640.jpg or https://www.researchgate.net/profile/Graziela-Heberle-2/publication/50265549/figure/fig7/AS:394356612714502@1471033351621/Fig-10-Superposition-of-the-best-docked-structure-and-crystallographic-structure-for_W640.jpg) . I am not convinced that with the current way of reporting the docking proves anything.

Response: The authors are extremely thankful to the reviewer for sparing valuable time and for such a very professional review. Also, thank you so much for your valuable insight on improving the manuscript. We greatly appreciate the reviewers’ comments and constructive suggestions. This valuable comment is constructive to improve our manuscript and provide meaningful guidance for our future research. The authors have done validation of docking according to the suggestion of the reviewer and is highlighted in the manuscript. We hope this revised docking validation will meet the satisfaction of the reviewer. We again appreciate your helpful suggestions.

Round 3

Reviewer 2 Report

General comments:

1. Again, standardize naming in tables and paragraphs! Once you decide whether you write with a hyphen and with a capital or lowercase letter, don't change.  

2. Correct the references, they still contain errors. 

3. The paragraph in lines 64-72 needs to be corrected. There are numerous repetitions in close proximity.

Detailed comments:

Line 126.          Maybe better would be the usage of minus sign?

Line 156-158    This sentence must be improve.

Line 238.         Value with lowercase letter

Line 413.         Should be 'absorption'

Figure 3            F. vasta should be in italic

Author Response

Response to Respected Reviewer 2 comments

Thank you for your kind observation. Also, thank you so much for your valuable insight on improving the manuscript. We greatly appreciate the reviewers’ comments and constructive suggestions. The reference section was also improved accordingly. The research design and results are clearly presented. We hope this revised manuscript will meet the satisfaction of the reviewer. We again appreciate your helpful suggestions. If you have any further suggestions for changes, please let us know.

  1. Again, standardize naming in tables and paragraphs! Once you decide whether you write with a hyphen and with a capital or lowercase letter, don't change. 

Response: The authors are thankful for such a professional review. The uniformity is made in the manuscript.

  1. Correct the references, they still contain errors.

Response: The authors have made changes in the reference section manually.

  1. The paragraph in lines 64-72 needs to be corrected. There are numerous repetitions in close proximity.

Response: This paragraph is now rephrased and repetitions are now corrected.

Line 126.          Maybe better would be the usage of minus sign?

Line 156-158    This sentence must be improve.

Line 238.         Value with lowercase letter

Line 413.         Should be 'absorption'

Figure 3            F. vasta should be in italic

Response: The authors highly acknowledge all the comments and now all the described changes have been done and highlighted.

Reviewer 4 Report

The reporting on the docking validation has improved, but now there are two Figures S3 and the authors need to correct it. Besides, they mention that RMSD values should be shown on the new Figure S3, but no RMSD value seems to be reported now. These two aspects will have to be solved.

Author Response

Response to Respected Reviewer 4 comments

Response: Thank you for your kind observation. Also, thank you so much for your valuable insight on improving the manuscript. We greatly appreciate the reviewers’ comments and constructive suggestions. Those valuable comments are constructive to improve our manuscript and provide meaningful guidance for our future research. We hope this revised manuscript will meet the satisfaction of the reviewer. We again appreciate your helpful suggestions. If you have any further suggestions for changes, please let us know.

The reporting on the docking validation has improved, but now there are two Figures S3 and the authors need to correct it. Besides, they mention that RMSD values should be shown on the new Figure S3, but no RMSD value seems to be reported now. These two aspects will have to be solved.

Response: The authors acknowledge reviewer comments. The Figure numbering in the supplementary file and main text has been corrected. The RMSD values are mentioned in the manuscript and highlighted.